# Plant water resource partitioning and isotopic fractionation during transpiration in a seasonally dry tropical climate

Lien De Wispelaere[1], Samuel Bodé[1], Pedro Hervé-Fernández[1,2], Andreas Hemp[3], Dirk Verschuren[4*] and Pascal Boeckx[1*]

[1]Isotope Bioscience Laboratory – ISOFYS, Faculty of Bioscience Engineering, Ghent University, Coupure links 653, 9000 Gent, Belgium
[2]Laboratory of Hydrology and Water Management, Faculty of Bioscience Engineering, Ghent University, Coupure links 653, 9000 Gent, Belgium
[3]Department of Plant Systematics, University of Bayreuth, 95440 Bayreuth, Germany
[4]Limnology Unit, Department of Biology, Gent University, Ledeganckstraat 35, 9000 Gent, Belgium

*Correspondence to*: Lien De Wispelaere (lien.dewispelaere@ugent.be)

* These authors contributed equally to this work

**Abstract.** Lake Challa (3°19'S, 37°42'E) is a steep-sided crater lake situated in equatorial East Africa, a tropical semi-arid area with bimodal rainfall pattern. Plants in this region are exposed to a prolonged dry season and we investigated if 1) these plants show spatial variability and temporal shifts in their water source use; 2) seasonal differences in the isotopic composition of precipitation are reflected in xylem water; and 3) plant family, growth form, leaf phenology, habitat and season influence the xylem-to-leaf water deuterium enrichment. In this study, the $\delta^2H$ and $\delta^{18}O$ of precipitation, lake water, groundwater, plant xylem water and plant leaf water were measured across different plant species, seasons and plant habitats in the vicinity of Lake Challa. We found that plants rely mostly on water from the 'short' rains falling from October to December (northeastern monsoon), as these recharge the soil after the long dry season. This plant-available, static, water pool is only slightly replenished by the 'long' rains falling from February to May (southeastern monsoon), in agreement with the 'two water world' hypothesis according to which plants rely on a static water pool while a mobile water pool recharges the groundwater. Spatial variability in water resource use exists in the study region with plants at the lake shore relying on a water source admixed with lake water. Leaf phenology does not affect water resource use. According to our results, plant species and their associated leaf phenology are the primary factors influencing the enrichment in deuterium from xylem water to leaf water ($\varepsilon_{l/x}$), with deciduous species giving the highest enrichment; while growth form and season have negligible effects. Our observations have important implications for the interpretation of $\delta^2H$ of plant leaf wax *n*-alkanes ($\delta^2H_{wax}$) from paleohydrological records in tropical East Africa, given that the temporal variability in the isotopic composition of precipitation is not reflected in xylem water and that leaf water deuterium enrichment is a key factor in shaping $\delta^2H_{wax}$. The large interspecies variability in xylem-leaf enrichment ($24 \pm 28$ ‰) is potentially troublesome, taking into account the likelihood of changes in species assemblage with climate shifts.

## 1 Introduction

The naturally occurring stable isotopes of hydrogen and oxygen in the water molecule have been highly instructive as tracers in hydrology and eco-hydrology. This is mainly based on naturally occurring variations in the relative abundance of two rare, heavy isotopes (i.e. $^2$H or D and $^{18}$O), arising from phase changes and mixing as water passes through the hydrologic cycle (Dansgaard, 1964; Gat 1996). The hydrogen and oxygen isotopic composition of precipitation varies both spatially and temporally, due to fluctuations i) at the site of evaporation,

e.g. in meteorological conditions such as relative humidity (RH), wind and sea surface temperature; and ii) at the site of precipitation, e.g. in the degree of rainout of particular air masses (Craig, 1961; Dansgaard, 1964; Gat 1996; Araguas-Araguas et al., 2000, Gibson et al., 2008). The stable isotopes of hydrogen and oxygen in precipitation show a distinct empirical relationship, described by the Global Meteoric Water Line (GMWL: $\delta\,^2$H = 8.1* $\delta\,^{18}$O + 10.3 ‰; Rozanski et al., 1993). The $\delta\,^{18}$O- $\delta\,^2$H relationship in precipitation at any single location

is however better described by a Local Meteoric Water Line (LMWL), which can have a different slope and intercept depending on the conditions in which the local water source was formed, and LMWLs can be used to compare different (sub)surface water bodies with local precipitation (Rozanski et al., 1993; Breitenbach et al., 2010).

Variation of $\delta\,^2$H or $\delta\,^{18}$O in xylem water of plants has been used extensively to determine water sources used

by plants and their functional rooting depth (Zimmermann et al., 1967; Brooks et al., 2010). Evaporation directly from the soil causes an isotopic enrichment of soil water available for plant roots (Sachse et al., 2012 and references therein). Water uptake through the roots and transport in plants is generally considered to occur without fractionation (White et al., 1985; Dawson and Ehleringer, 1991, 1993; Zhao et al., 2016), so that the isotopic composition of xylem water represents the composition of the plant water source (Dawson & Ehleringer,

1991; Evaristo et al., 2015). Fractionation during root water uptake has been found for plants living under xeric conditions and in mangroves (Ellsworth & Williams, 2007) and recent research stated that fractionation likely occurs during the water redistribution after uptake (Zhao et al., 2016). In contrast, the isotopic composition of leaf water differs markedly from that of xylem water. This is because evaporation discriminates towards lighter isotopologues. As a result, the remaining leaf water after transpiration becomes more enriched in heavy isotopes.

The degree of enrichment from xylem to leaf water is a function of temperature, RH and the isotopic composition of water vapor surrounding the plant (Kahmen et al., 2008; Sachse et al., 2012).

As δ²H values of leaf wax $n$-alkanes obtained from lake surface sediments and soils showed tight correlations with δ²H values of precipitation (Sachse et al., 2004; Hou et al., 2008; Polissar and Freeman, 2010; Garcin et al., 2012), the δ²H signature of $n$-alkanes, long-chain hydrocarbons with 25-35 carbon atoms derived from fossil

plant leaf waxes incorporated in lake sediments, is increasingly being used as paleoclimate proxy (Eglinton and Hamilton, 1967; Mayer and Schwark, 1999; von Grafenstein et al., 1999; Thompson et al., 2003; Tierney et al. 2008; Costa et al. 2014). Consequently, also a better understanding of the hydrogen fractionation is needed which occurs during its incorporation from precipitation via leaf water into plant leaf waxes. The present study is developed in the context of such an application of hydrogen isotope geochemistry for paleoclimate research,

focusing on the sediment record of Lake Challa (Verschuren et al. 2009; Barker et al., 2011; Tierney et al., 2011).

The study area is located in equatorial East Africa. In this semi-arid tropical region, biannual passage of the Intertropical Convergence Zone (ITCZ) induces a strongly bimodal pattern of seasonal rainfall (Nicholson, 2000). Plants in this region are exposed to a prolonged dry season between June and September during which little precipitation falls (<20 mm month$^{-1}$). Adaptations to survive this period of water shortage include stem succulence, leathery leaves and deep roots providing access to deep and permanent water sources (Elffers et al., 1964; Corbin et al., 2005). Meinzer et al. (1999) suggested that, at least in pristine dryland ecosystems, competition for water among species may actually be limited due to pronounced spatial and temporal partitioning of water resources resulting from maximized species diversity. Differential water resource utilization has been shown across different (Ehleringer et al., 1991; Jackson et al., 1999; Goldstein et al., 2008) and within similar growth forms (Field and Dawson, 1998; Meinzer et al., 1999; Stratton et al., 2000). It furthermore appears that the relationship between the root biomass in a particular soil layer and the degree of contribution of that soil layer to plant water uptake is not always straightforward (Jackson et al., 1995; Stahl et al., 2013). Therefore, analysis of the dual stable isotope composition of xylem water could be a valuable tool to elucidate the water sources effectively used by plants (Dawson & Ehleringer, 1991; Liu et al., 2010).

In this study, we measured the $\delta$ $^2$H and $\delta$ $^{18}$O content of precipitation, lake water, groundwater, and of xylem and leaf water in 3 individuals of each of the 14 different species from three distinct habitats around Lake Challa. Sampling was carried out during successive wet and dry seasons of one complete year. Our primary research questions were: 1) are seasonal differences in the isotopic composition of precipitation reflected in xylem water? 2) do plants show habitat-specific variability and temporal shifts in their water source use? and (3) what is the influence of plant family, growth form, phenology, season and habitat on the deuterium enrichment from xylem to leaf water? The work that we present here reports isotope data from a data scarce region and is intended to provide a basis for the interpretation of leaf wax $n$-alkane δ²H values applied as (paleo)hydrological proxies.

## 2 Materials & methods

### 2.1 Study site

Lake Challa is a 4.2 km², ~92 m deep crater lake with near-vertical inner crater walls (Moernaut et al., 2010), situated on the southeastern slope of Mt. Kilimanjaro (3°19'S, 37°42'E) at 880 m above sea level (m.a.s.l.) in equatorial East Africa (Fig. 1). Biannual passage of the Intertropical Convergence Zone (ITCZ) induces a bimodal rainfall pattern with southeastern (SE) monsoon winds bringing 'long rains' normally from March to mid-May and northeastern (NE) monsoon winds bringing 'short rains' from late October to December (Nicholson, 2000; Fig. 2). The local climate is tropical semi-arid, with lowest mean monthly night- and daytime temperatures in July-August (c. 18 and 28 °C, respectively) and highest values in February-March (c. 21 and 33 °C, respectively; Fig. 1a) for Voi, 80 km east of Lake Challa (Buckles et al., 2014). Given the total annual rainfall of c. 565 mm and an estimated annual lake-surface evaporation of c. 1735 mm (Payne, 1970), the water budget of Lake Challa must be balanced by substantial groundwater input, which main source is precipitation falling onto montane forests of Mt. Kilimanjaro's east-facing slope at 1800-2800 m.a.s.l. (Hemp, 2006a).

The vegetation of the crater basin containing Lake Challa consists of different forest and woodland types (Hemp, 2006b). On the upper part of the inner slopes a dry forest occurs, with succulents such as *Euphorbia quinquecostata* (Euphorbiaceae) and deciduous species such as *Commiphora baluensis* (Burseraceae) and *Haplocoelum foliolosum* (Sapindaceae) dominating the tree layer. Near the lake shore an evergreen forest with *Sorindeia madagascariensis* (Anacardiaceae), *Ficus sycomorus* (Moraceae) and *Trichilia emetica* (Meliaceae) grows. In contrast, the outer crater slopes are covered with dry savanna woodlands, with a lower and more open canopy. The stunted, fruit tree-like appearance of the woody species, mainly Combretaceae, Burseraceae and Anacardiaceae, inspired the first botanists to describe this vegetation formation as 'Obstgartensteppe' ('fruittreegardensteppe'; Volkens, 1897). Whereas all these vegetation types grow on rocky slopes with very shallow soils, the soils of the flat foothills are deeper. Here, most of the former natural savanna vegetation is converted into agricultural fields or meadows. The savanna woodlands still existing in this area are dominated by Acacias (*A. nilotica*, *A. senegal*; Mimosaceae).

**2.2 Sampling**

Two duplicate rain gauges were installed in the savannah just outside Challa crater (at 3°19'S 37°42'E and 842 m.a.s.l.; Fig. 1) to sample precipitation monthly between September 2013 and August 2014. The collectors consisted of a 5 L plastic container with a plastic funnel of c. 15 cm diameter, in which a plastic fiber mesh net was placed to prevent dirt from entering the bottle. A layer of mineral oil (thickness ca. 1.5 cm) was poured in the jars to avoid evaporation and exchange with air moisture, which could alter isotopic composition of collected water (Friedman et al., 1992). As only 6 precipitation samples reached the laboratory, precipitation was additionally collected in the savannah about 20 km to the west of the Challa crater (at 3°23'S 37°27'E and 820 m.a.s.l.) by an evaporation-free collector (Groning et al., 2012) of the brand PALMEX (Croatia) on a monthly basis from November 2014 until November 2015. Lake water was sampled 30 cm below the water surface in the middle of the lake on a monthly basis from January 2013 until October 2014. A groundwater sample was obtained from the Miwaleni spring (3°25'S 37°27'E) in July 2015. We assumed that the isotopic composition of groundwater is seasonally stable, as the groundwater recharge occurs at decadal scale in the region (Zuber, 1983).

Plant material was collected during the main dry season in September 2013 and July 2014, during the NE monsoon season in December 2013, during the short dry season in February 2014 and during the SE monsoon season in April 2014. However, because the 2014 long rainy season atypically started already in February, our plant sampling for the short dry season was already influenced by fresh SE monsoon rainfall. Fourteen plant species with varying growth form (grass, shrub or tree) and leaf phenology (deciduous or evergreen, the latter including succulent) were collected in three distinct habitats (savannah, crater rim, lake shore), representative for the region, around Lake Challa (Fig. 1, Table 1). Shrubs were defined as woody plants with multiple stems, while trees had one erect perennial stem. Lakeshore vegetation was sampled at the northeast side of the lake, savannah was sampled outside the crater c. 500 m to the northwest, and crater-rim vegetation at the top of the crater's western rim (1100 m.a.s.l.). The locally most abundant plant species within each habitat were chosen with the aid of an experienced local guide, although the choice of species was sometimes restricted by practical

limitations such as difficulties in reaching certain locations. For each habitat, three individuals of each species were sampled.

Two different sampling techniques were used to collect xylem water from plants, the choice of which depended on the plant type. When the plant had a trunk with a diameter of more than 10 cm, a core drill sample (300 mm, diameter 4.30 mm, hard-wood head, Pressler, Recklinghausen, Germany) was extracted, from which the outer layer (epidermis, cortex, bark fibres, and phloem) was removed to prevent contamination with phloem sap. In the case of smaller trees and shrubs, a piece of twig was sampled, the outer layer was scraped off using a knife and it was enclosed into sealed vials. For leaf water analysis, leaves were taken from each plant and placed in vials. If twigs were sampled, leaves on those twigs were sampled. In case of core sampling, leaves were randomly sampled at different heights and at the four cardinal points, and merged to one bulk sample per replicate to provide samples that were representative for the entire plant. Entire leaves were collected in order to ensure integration of the signal from the entire leaf, given likely isotopic gradients along the length of the leaf (Helliker and Ehleringer, 2000; Sessions, 2006). Leaves were sampled between 10 a.m. and 3 p.m. to eliminate additional variability induced by previously reported large diurnal variations in the isotopic composition of leaves (Cernusak et al., 2002; Li et al., 2006; Kahmen et al., 2008). From grasses, only leaves were sampled. The whole plant was sampled which consisted mainly of green leaves and thus represented leaf water. Stem, twig, and leaf samples were stored frozen until the water was quantitatively extracted via cryogenic vacuum distillation (West et al., 2006). Following Araguàs-Araguàs et al. (1995), isotopic data was retained for interpretation only if the extraction efficiency, determined by further drying of the sample at 105°C for at least 48 hours, exceeded 98%. A variety of water extraction methods for the analysis of the stable isotopes water exist. Cryogenic vacuum extraction of soil pore water remains a challenge, but is an effective method for plant water extractions (Orlowski et al., 2016).

**2.3 Analysis**

The $\delta^2H$ and $\delta^{18}O$ values of water samples were determined using Cavity Ringdown Spectrometry (WS-CRDS, L2120-i, Picarro, USA), coupled with a vaporizing module (A0211 high-precision vaporizer) and a microcombustion module, which eliminates interference of organic compounds (Martín-Goméz et al., 2015). Each sample was measured 10 times, of which the first 5 injections were eliminated in order to overcome memory effects. The measurement uncertainty ($\pm$ 1$\sigma$) of our CRDS was 0.1 ‰ and 0.4 ‰ for $\delta^{18}O$ and $\delta^2H$, respectively. Isotopic composition is expressed in terms of $^2H/^1H$ and $^{18}O/^{16}O$ ratios, represented by $\delta$ values: $\delta_{sample} = (R_{sample}/R_{standard} - 1)$ with $R_{sample}$ and $R_{standard}$ being the isotopic ratio ($^2H/^1H$ or $^{18}O/^{16}O$) measured in the sample and the standard, respectively (Gat, 2005). The used reference standard is Vienna Standard Mean Ocean Water (VSMOW) which, by definition, has $\delta^2H$ and $\delta^{18}O$ concentrations of 0 ‰.

The enrichment factor $\varepsilon_{l/x}$ characterizes the hydrogen-isotopic fractionation between xylem and leaf water and is defined as (Eq. 1):

$$\varepsilon_{l/x} = ((\delta^2H_{leaf} + 1)/(\delta^2H_{xylem} + 1)) - 1 \tag{1}$$

Enrichment factors and $\delta$ values are typically reported in per mil (‰) (Cohen et al., 2007). Both the $\delta^2H$ and $\delta^{18}O$ values of water samples were measured, but particular focus was given on the $\delta^2H$ values as this research is intended to provide a basis for the interpretation of leaf wax *n*-alkane $\delta^2H$ values as (paleo)hydrological proxies.

 The average isotopic signature of the source of xylem water was determined from the intersection of xylem water samples (aligned along a local evaporation line, LEL) with the LMWL (Eq. 2 and 3):

$$\delta^{18}O_{LMWL-int} = \frac{\delta^2H - slope_{LEL} * \delta^{18}O - intercept_{LMWL}}{slope_{LMWL} - slope_{LEL}} \tag{2}$$

$$\delta^2H_{LMWL-int} = \delta^{18}O_{LMWL-int} * slope_{LMWL} + intercept_{LMWL} \tag{3}$$

The isotopic signatures of xylem water were further characterized with a parameter describing the relative degree of evaporation. We developed the evaporation distance, defined as ED and calculated as the distance from the LMWL along an evaporation line, scaled to the $\delta^2H$ axis (Eq. 4). The higher this ED value, the further away from the LMWL and the more evaporated the water will be, i.e. the higher the concentration of heavy isotopes.

$$\text{Evaporation distance (ED)} = \sqrt{(\delta^2H - \delta^2H_{LMWL-int})^2 + slope_{LMWL} * \left(\delta^{18}O - \delta^{18}O_{LMWL-int}\right)^2} \tag{4}$$

Analyses of variance (ANOVA) were used for comparisons of $\delta^2H$ and $\delta^{18}O$ isotopic signatures among plant species, growth forms, leaf phenologies, seasons and habitats. Tukey post-hoc comparisons were used to further examine differences. All statistical analyses were performed using R (version 3.2.3.). Slopes and intercepts of LMWL and LEL were estimated with linear regressions. A discussion of the different slopes and intercepts is not  the scope of this paper, but they were used to calculate $\delta^2H_{LMWL-int}$.

## 3 Results

### 3.1 Isotopic composition of precipitation, lake water and groundwater

During our main sampling period from September 2013 until August 2014, local rainfall was highest in  December (peak NE monsoon rains) and April (peak SE monsoon rains) with 135 and 122 mm, respectively, and lowest in January (short dry season) and August (long dry season) with two times 1 mm of rain (Fig. 2b). During the 12 month monitoring period, the total amount of local precipitation was 692 mm. The monthly average temperature varied between 23.9 °C in June-July and 27.1 °C in April, resulting in an overall annual mean of 25.5 ± 1.2 °C (mean ± 1 $\sigma$ standard deviation). The monthly minimum (night-time) temperature followed a  similar pattern (19.6 ± 1.0 °C), while the monthly maximum (day-time) temperature showed greater variability (37.5 ± 2.1 °C), with an atypical minimum in February (34.8 °C) and maximum in April (41.2 °C).

The isotopic composition of precipitation is most enriched during the dry month of July with values of 36.6 for $\delta^2H_{prec}$ and 4.2 ‰ for $\delta^{18}O_{prec}$ and most depleted during rainy November with values of -47.9 and -7.2 ‰ (Fig. 2), respectively. Comparing the two rainy seasons revealed considerable differences (p = 0.08) with more enriched rain during the SE monsoon ($\delta^2H_{prec}$ of 16.0 ± 2.5 ‰ and $\delta^{18}O_{prec}$ of 0.8 ± 0.7 ‰; n = 3) compared to the NE monsoon ($\delta^2H_{prec}$ of -26.5 ± 21.5 ‰ and $\delta^{18}O_{prec}$ of -4.9 ± 2.3 ‰; n = 2). In order to draw a LMWL, precipitation samples (n = 12) covering the period November 2014 - November 2015 were added (from a savannah site 20 km west of Lake Challa, cf. above). Based on this dataset (n = 12), the yearly volume-weighted average values are -6.5 ‰ for $\delta^2H_{prec}$ and -2.5 ‰ for $\delta^{18}O_{prec}$. Compared to the global meteoric water line ($\delta^2H = 8.1*\delta^{18}O + 10.3$ ‰, Rozanski et al., 1993) and the LMWL of central Kenya ($\delta^2H = 8.3*\delta^{18}O + 11.0$ ‰, Soderberg et al., 2013), the LMWL of the study region ($\delta^2H = 7.1*\delta^{18}O + 10.7$ ‰, n = 18) has a slightly lower slope and intermediate intercept (Fig. 3).

The isotopic measurements on lake water during 22 consecutive months (from January 2013 until October 2014) yielded mean $\delta^2H_{lake}$ and $\delta^{18}O_{lake}$ values of 17.4 ± 0.7 ‰ and 2.9 ± 0.2 ‰ respectively (Fig. 2 and 3), with very little variation through the year but highest values in the warm months of February-March and a modest minimum around August. The groundwater isotopic composition equaled -20.2 ‰ for $\delta^2H$ and -4.6 ‰ for $\delta^{18}O$ in July 2015 (Fig. 3).

**3.2 Xylem water**

The $\delta^2H$ of xylem water ($\delta^2H_{xylem}$) in a total of 154 analyzed samples (no grasses) ranged between -87 and 25 ‰ (Fig. 3), with an overall mean value of -18 ± 17 ‰. $\delta^2H_{xylem}$ varied between plants at the lake shore (-2 ± 10 ‰, n = 48) and isotopically more depleted plants in the savannah (-25 ± 12 ‰, p <0.01, n = 34) and on the crater rim (-26 ± 15 ‰, p <0.001, n = 72). The $\delta^2H_{xylem}$ of trees (3 species, n = 38) at the lake shore (1 ± 8 ‰) was significantly higher than that of the single shrub species sampled in this habitat (-13 ± 5 ‰, p <0.001, n = 10). In the savannah and on the crater rim, no difference (p >0.05) could be observed between the $\delta^2H_{xylem}$ of trees and shrubs. Across all sampled plants, leaf phenology (deciduous or evergreen) did not significantly influence $\delta^2H_{xylem}$ value (p >0.05). Only two of the sampled species showed seasonality in $\delta^2H_{xylem}$, but in a dissimilar pattern. The tree species *Sideroxylon* sp. had lower $\delta^2H_{xylem}$ values during the long dry season (0 ± 5 ‰, p <0.05) than during the short (10 ± 1 ‰) and long rainy seasons (6 ± 8 ‰). The tree species *Ficus sycomorus* showed lower values (p < 0.01) during the short rainy season (-5 ± 2 ‰) than during the long dry season (4 ± 3 ‰) and long rainy season (13 ± 4 ‰).

The hydrogen isotopic signatures of xylem samples follow an evaporation line (LEL). To determine the mean isotopic composition of the water source from which xylem water originated LELs were calculated for each of the three different plant habitats and used to estimate (Eq. 2 and 3) the intersection points of xylem water with the LMWL ($\delta^2H_{LMWL-int}$). LELs with a slope of about 3 fitted best with our data from savannah and the crater rim, while a slope of about 5 fitted best at the lake shore. The LELs with slope of 3 correspond well with the modeled evaporation lines for soil water in our study area (Gibson et al. 2008). A slope of 5 corresponds more

with these authors' modeled evaporation lines for surface water, which can be explained by the lake shore trees and shrubs mostly using lake water. The $\delta\,^2H_{LMWL\text{-}int}$ values ranged between -79 and -13 ‰, with an overall mean value of -41 $\pm$ 13 ‰. No statistical differences (p >0.05) could be observed among $\delta\,^2H_{LMWL\text{-}int}$ values analyzed by habitat, species, growth form or leaf phenology (Fig. S1). Plants at the lakeshore showed only a weak temporal trend in $\delta\,^2H_{LMWL\text{-}int}$ (mean -45 $\pm$ 12 ‰, p >0.05), whereas plants in both the savannah (-42 $\pm$ 9 ‰, p <0.01) and on the crater rim (-38 $\pm$ 15 ‰, p <0.001) showed significant seasonal variability in $\delta\,^2H_{LMWL\text{-}int}$ (Fig. 4).

The evaporation distance (ED, Eq. 4) is a parameter describing the relative degree of evaporation of a xylem water sample. It is derived by calculating the distance of xylem data points from the LMWL along the LEL. The higher this ED value, the further away from the LMWL and the more evaporated the water will be. A great range in ED values was observed, varying between 1 and 94 ‰ across all samples (Fig. 5). Plants at the lakeshore produced systematically higher ED values (49 $\pm$ 13 ‰) than those in savannah (23 $\pm$ 14 ‰, p <0.001) or on the crater rim (17 $\pm$ 9 ‰, p <0.001). Growth form also influenced ED (p <0.05) with lower values for shrubs than for trees both at the lake shore (respectively 42 $\pm$ 16 ‰ and 50 $\pm$ 12 ‰) and on the crater rim (respectively 16 $\pm$ 9 ‰ and 22 $\pm$ 8 ‰, Fig. S2). No difference (p >0.05) in ED value was found between evergreen and deciduous plants. The evaporation distance showed a clear temporal effect during the study period at the lake shore (p <0.01) and on the crater rim (p <0.01), but in the savannah the trend was not significant (p >0.05, Fig. 5). Among the seven non-grass plant species sampled at the crater rim, a significant difference (p <0.01) was observed between the low ED value for *Vepris uguenensis* (9 $\pm$ 7 ‰) and the high ED value for *Euphorbia tirucalli* (25 $\pm$ 7 ‰).

### 3.3 Leaf water

The number of samples investigated for leaf water $\delta\,^2H$ ($\delta\,^2H_{leaf}$) totaled 186 including the two species of grasses. Across this complete dataset $\delta\,^2H_{leaf}$ ranged from -83 to 37 ‰ with an overall mean value of 5 $\pm$ 20 ‰ (Fig. 3). The $\delta\,^2H_{leaf}$ of grasses (mean -2 $\pm$ 14 ‰, n = 18) were less enriched than those of trees (10 $\pm$ 16 ‰, p <0.01, n = 63), while those of shrubs were intermediate (3 $\pm$ 22 ‰, p = 105). Leaf phenology had a significant effect (p <0.001) on the $\delta\,^2H_{leaf}$ of shrubs with values of 16 $\pm$ 7 ‰ and -4 $\pm$ 24 ‰ for deciduous (n = 38) and evergreen (n = 67) shrubs, respectively. The two evergreen shrubs of the Capparaceae family in particular showed strongly depleted $\delta\,^2H_{leaf}$ values, respectively -11 $\pm$ 19 ‰ for *Thylachium africanum* and -57 $\pm$ 14 ‰ for *Maerua* sp. Even if these outliers are removed from the group of evergreen shrubs, the difference in $\delta\,^2H_{leaf}$ between deciduous and evergreen remains significant (p < 0.05). No such difference (p >0.05) was observed between the $\delta\,^2H_{leaf}$ of deciduous (n = 23) and evergreen (n = 40) trees. At the lake shore, trees showed higher $\delta\,^2H_{leaf}$ values (9 $\pm$ 20 ‰, n = 38) than shrubs (-6 $\pm$ 20 ‰, p <0.05, n = 12). Shrubs in the savannah had higher $\delta\,^2H_{leaf}$ values (13 $\pm$ 13 ‰, n = 31) than those on the crater rim (0 $\pm$ 25 ‰, p <0.05, n = 62) and at the lake shore (-6 $\pm$ 20 ‰, p <0.05, n = 12), while $\delta\,^2H_{leaf}$ values of shrubs at the lake shore and on the crater rim did not differ significantly (p >0.05). Only the shrub *Vepris uguenensis* (in both habitats) and the trees *Sideroxylon* sp. and *Lepisanthes senegalensis* (only lakeshore habitat) showed an effect of seasonality on $\delta\,^2H_{leaf}$. *V. uguenensis* showed depleted $\delta\,^2H_{leaf}$ values during the dry season (-11 $\pm$ 16 ‰) versus enriched values during both rainy

seasons (7 ± 11 ‰), whereas *Sideroxylon* sp. and *L. senegalensis* showed depleted $\delta\,^2H_{leaf}$ during the long rainy season (-10 ± 20 ‰ and -6 ± 11 ‰ respectively) and enriched $\delta\,^2H_{leaf}$ during the short rainy season (27 ± 3 ‰ and 33 ± 3 ‰, respectively).

**3.4 Factor of deuterium enrichment from xylem to leaf water (e$_{l/x}$)**

The enrichment factor $\varepsilon_{l/x}$ of deuterium fractionation between xylem and leaf water could be determined on a total of 133 pairs of $\delta\,^2H_{xylem}$ and $\delta\,^2H_{leaf}$ values. This yielded an average $\varepsilon_{l/x}$ for $\delta\,^2H$ of 24 ± 28 ‰ across all habitats, plant species (trees and shrubs only) and seasons (Fig. 6); the $\varepsilon_{l/x}$ for $\delta\,^{18}O$ is not reported here but showed the same trends. The enrichment factor showed a significant difference between plants at the lake shore (7 ± 23 ‰, p <0.001, n = 42) and the savannah (36 ± 19 ‰, p <0.001, n = 29) and crater-rim plants (30 ± 30 ‰, n = 62). Growth form had no significant effect (p >0.05) on $\varepsilon_{l/x}$ with values of 27 ± 29 ‰ for shrubs (n = 80) and 19 ± 25 ‰ for trees (n = 53). Significant differences (p <0.001) were found between species according to their leaf phenology with $\varepsilon_{l/x}$ values of 37 ± 25 ‰ deciduous plants (n = 50) and 16 ± 27 ‰ for evergreens (n = 83). An effect of seasonality was limited and could only be observed in *Sideroxylon* sp. (p <0.05) and *Lepisanthes senegalensis* (p <0.01), reflecting the trends in $\delta\,^2H_{leaf}$.

**4 Discussion**

**4.1 Water sources: isotopic composition of precipitation, groundwater and lake water**

Equatorial East Africa has a pronounced bimodal seasonality in rainfall, characterized by '(long) SE monsoon rains' from March until May and '(short) NE monsoon rains' from late October until December separated by a long dry season (Nicholson, 2000). During November-December of 2013, when Indian Ocean moisture was advected by NE monsoon winds, $\delta\,^2H_{prec}$ and $\delta\,^{18}O_{prec}$ were more depleted than during February through May 2014, when Indian Ocean moisture was advected by SE monsoon winds (Fig. 2). This result stresses the importance of the air mass trajectory in controlling seasonal patterns of rainwater isotopic signatures. For the location of Lake Challa, moist air advected by the NE monsoon has travelled a longer distance overland compared to moist air advected by the SE monsoon. However, within the short rainy season we recorded a considerable difference between the very strongly depleted $\delta\,^2H_{prec}$ value for November precipitation (-48 ‰), representing the first rains after the dry season, and only modestly depleted $\delta\,^2H_{prec}$ value for December (-5 ‰). This indicates that not only the general air mass trajectory but also other phenomena such as different degrees of rainout contributing to the formation of precipitation or temperature and relative humidity control of the initial vapour (Dansgaard, 1964) play an important role in determining the isotopic composition of monthly precipitation in any particular year. The Hybrid Single Particle Lagrangian Integrated Trajectory (HYSPLIT) model (Draxler and Hess 2004), developed by the National Oceanic and Atmospheric Administration (NOAA), confirmed that there is a distinctly different trajectory for precipitation in November and December (northeast) and April, May and July (southeast). The total amount of rainfall during the study period (692 mm) was slightly above reported values for the long-term mean annual precipitation in the Challa region, which vary between 583

mm for Taveta 1989-2005 and 532 mm for Challa 2000-2007 (Fig. 2) and ~650 mm for Challa region (Hemp, 2006b). According to the Kenya Food Security Steering Group (2014), the 2013 rainy season started in mid-November instead of late October and was thus delayed by 2-3 weeks. Additionally, it already ceased in mid-December, a week earlier than normal. Rainfall amounts in both November and December were well above average. In addition, the 2014 long rainy season started earlier and ceased later than normal, with monthly rainfall amounts for February and June more than double those of an average year (Fig. 2). On the other hand, the month of January 2014 was exceptionally dry, and the main dry season from July to September was drier than usual (Fig. 2). The HYSPLIT model (Draxler and Hess 2004), developed by NOAA, confirmed that there is a distinctly different trajectory for precipitation in November and December (northeast) and April, May and July (southeast). To compute air parcel trajectories, the model required data from the NOAA meteorological database, and trajectories were modeled 310 hours backwards in time starting from the end of the respective month. The $\delta\ ^2H_{prec}$ and $\delta\ ^{18}O_{prec}$ in the dry month of July were clearly more enriched than the corresponding values for both wet periods (Fig. 2). This result illustrates the 'amount effect' (Dansgaard, 1964), which states that tropical regions characterized by limited temperature variation but strong seasonality in rainfall show a stronger depletion of the heavy isotopes of water at higher precipitation rates.

The isotopic composition of the groundwater sampled at Miwaleni spring in July 2015 (-4.6 ‰ for $\delta\ ^{18}O$ and -20.2 ‰ for $\delta\ ^2H$) is consistent with data of McKenzie et al. (2011) for several groundwater wells around Mt. Kilimanjaro measured in 2006.

Monthly isotopic measurements of lake-surface water revealed very little seasonal variation in $\delta\ ^{18}O_{lake}$ and $\delta\ ^2H_{lake}$ (Fig. 2 and 3). Our mean $\delta\ ^2H_{lake}$ and $\delta\ ^{18}O_{lake}$ values (17.4 ‰ and 2.9 ‰ respectively) during 2013-2014 are very similar to those measured by McKenzie et al. (2010) in January 2006 (19.5 ‰ and 2.3 ‰), indicating that the isotopic signature of Lake Challa surface water is also stable on an inter-annual to decadal time scale. We did observe a small seasonal trend in both $\delta\ ^{18}O_{lake}$ and $\delta\ ^2H_{lake}$ with lowest (least enriched) values around August. This is counterintuitive, as evaporative enrichment is expected to be more pronounced during the dry season. Although detailed assessment of this seasonal trend is outside the scope of the present study, we offer two possible explanations. First, seasonal deep circulation of the lake's water column during the cool and windy main dry season (June-September; Wolff et al., 2014) may mix the evaporating (and thus isotopically enriched) surface water with isotopically more depleted deeper water, resulting in slight depletion of the surface water. Alternatively, dry-season evaporation may be compensated by enhanced inflow of subsurface water carrying the isotopic signature of the precipitation which fell on the forested slopes of Mt. Kilimanjaro during the previous rainy season (Barker et al., 2011). Whatever the cause of the modest seasonal trend in lake-water $\delta\ ^2H_{lake}$, plants using significant amounts of lake water can be expected to show similarly small seasonality in the isotopic signature of xylem water, $\delta\ ^2H_{xylem}$, when compared to e.g. the seasonality of plants using temporarily more variable surface water.

**4.2 Xylem water: spatial and seasonal partitioning of plants' water sources**

The average isotopic composition of plants' source water is reflected in the intersection points of individual xylem samples' LELs with the LMWL. Plants that rely mostly on water from isotopically depleted NE monsoon rains will exhibit relatively low $\delta^2H_{LMWL\text{-}int}$ values, while the opposite is true for plants relying on water from the isotopically less depleted SE monsoon rains. The distance of individual $\delta^2H_{xylem}$ values from the LMWL along their LEL is proportional to the relative degree of evaporation before uptake by the plant. The higher this parameter ED, the greater the relative importance of topsoil water (which is prone to evaporation) compared to deeper soil water. In a study region experiencing a Mediterranean climate with wet winters and dry summers, Brooks et al. (2010) found increasingly depleted $\delta^{18}O$ and $\delta^2H$ values with increasing soil depth and argued that the first and isotopically depleted autumn rains recharged the deep and withered soil, whereas water in shallow soil contains water from later, more enriched precipitation. We did not measure the isotopic composition of water along a soil profile, but also in this study the first rains after the main dry season are isotopically the most depleted. Access of plants to groundwater will similarly result in isotopically depleted xylem water because this water is mainly derived from precipitation on the forests of Mt. Kilimanjaro at 1800-2800 m.a.s.l. (Payne, 1970; Hemp, 2006a). The distribution of precipitation on Mt. Kilimanjaro changes with elevation with mean annual precipitation (MAP) increasing with altitude along the southern slope to reach a maximum of ~2700 mm year$^{-1}$ at 2200 m.a.s.l. and decreasing rapidly further uphill (Hemp, 2001). Maximum groundwater recharge was found at an altitude of ~2,000 m where precipitation is depleted isotopically compared to Lake Challa due to the altitude effect (Dansgaard, 1964).

Whether plants in the Lake Challa area are deciduous or evergreen produced no systematic trends in $\delta^2H_{LMWL\text{-}int}$ and evaporation distance (Figs. S1-S2) at any of the three principal plant habitats. We expected that evergreen plants would be adapted to tap water from deep sources (i.e. isotopically depleted water), allowing them to survive the long dry season, as e.g. observed by Jackson et al. (1995) in a tropical moist lowland forest in Panama. However, in line with our results in a tropical dry lowland, Stratton et al. (2000) also failed to find a clear difference in $\delta^2H_{xylem}$ between deciduous and evergreen plants.

At the lake shore and on the crater rim, the evaporation distance of trees was higher than that of shrubs (Fig. S2), indicating that trees use more topsoil water enriched in heavy isotopes by evaporation. In contrast, in the savannah no such difference between the two growth forms was found. It is generally accepted that the deeper root systems of trees compared to shrubs allow them to access deeper soil water or groundwater (Dawson, 1996). However, Meinzer et al. (1999) however found smaller trees to use deeper sources of water than larger trees, and attributed this to three possible factors. Firstly, large trees require large amounts of nutrients to maintain their extensive crown leaf area, and the nutrient content of topsoil water is much greater than that of water taken from greater depth. Secondly, and likely related to the first factor, large trees have a relatively more extensive horizontal root system, in order to compensate for the reduced water content of the topsoil. Finally, the larger stem water storage capacity of large trees reduces peak daytime demands for soil water uptake and delay the onset of diurnal leaf water deficits. Goldsmith et al. (2011) agreed that soil nutrient availability can be a strong growth-limiting factor and therefore a driver of root distributions, but observed that plant species occurring in

either the understory or canopy of mature and secondary forests used a similarly shallow water source. In our study, the evaporation distances of plants (both trees and shrubs) at the lake shore were strikingly higher than that of plants growing in the savannah and on the crater rim (Fig. 5). This was because lake-shore plants use a substantial fraction of lake water, which has relatively enriched $\delta$ $^2$H$_{lake}$ values. The higher $\delta$ $^2$H$_{xylem}$ and evaporation distance of lake-shore trees compared to lake-shore shrubs indicate that the former rely more heavily on lake water while the latter tap more soil water (i.e. a smaller fraction of lake water).

Plants in all three local habitats showed similar, very negative $\delta$ $^2$H$_{LMWL-int}$ values, indicating that they relied mostly on depleted NE monsoon rains falling between October and December (Fig. 2). In all 3 habitats the $\delta$ $^2$H$_{LMWL-int}$ of plants sampled in December approached the $\delta$ $^2$H$_{prec}$ of November rain (-48 ‰; Fig. 4). Probably, this is because these rains represent the onset of the short rainy season following a distinct 4-month long dry season. They are thus expected to recharge soils to a relatively large degree. This is again in accordance with the 'two water worlds' (TWW) hypothesis of Brooks et al. (2010). This hypothesis challenges the assumption of water being completely mixed in soils and states that the mobile water compartment eventually enters the streams through translatory flow, while the static water compartment consists of initial precipitation that is trapped in soil micropores and remains trapped until the water is used through transpiration by plants in the following dry months (Brooks et al., 2010). Plants at the lake shore predictably showed only a weak (statistically insignificant) seasonal trend in $\delta$ $^2$H$_{LMWL-int}$ (Fig. 4). From observations it was obvious that the shore was the wettest of all three local habitats, and plants stayed verdantly green in the dry season as there was plenty of water available. Plants on the crater rim and in the savannah showed lowest $\delta$ $^2$H$_{LMWL-int}$ values during December, which then increased during the following months to reach their highest recorded values during the dry season in July. This indicates that the plants' water pool was replenished stepwise by the isotopically more enriched precipitation that followed from December onwards. At the top of the crater rim there was greater variability in monthly mean $\delta$ $^2$H$_{LMWL-int,}$ and the compound seasonal trend was more pronounced than in the savannah. Probably, this is because the rim is the driest location with shallow soils on bedrock, while the savannah site at the foot of the crater has deeper soils. Shallow soils have a smaller pool of micropores that can trap water and thus the static water compartment will be more quickly exhausted and replenished with new, isotopically different, water. Altogether, the isotopic composition of xylem water seems to confirm the 'two water world' hypothesis, in which a soil-bound water pool is used by the plants while another, highly mobile pool of precipitation water contributes to streams and groundwater (Brooks et al., 2010; Goldsmith et al., 2011; Evaristo et al., 2015).

From all plants on the crater rim, *Euphorbia tirucalli* had the highest evaporation distance, indicating that this evergreen shrub is adapted to use the most shallow water sources in this habitat. This allows it to exploit light precipitation events ('showers') more effectively (Caldwell et al., 1998) and to coexist with other species that draw water from deeper sources. This physiology may be due to the unique combination of CAM metabolism in the succulent stem of this species and C$_3$ metabolism in its non-succulent leaves (Van Damme, 1999; Hastilestari et al., 2013). *E. tirucalli* has a high drought tolerance as its leaves wither and die (i.e., become deciduous) under extremely dry conditions while the stem continues its CAM photosynthesis (Hastilestari et al., 2013). However,

the succulent stem made it difficult to separate phloem and xylem, so that the collected water may be a combination of both and its isotopic signature enriched because of transpiration.

**4.3 Parameters affecting xylem-to-leaf deuterium enrichment**

Averaging all sampled species over all seasons, the $\delta^2$H enrichment factor $\varepsilon_{l/x}$ from xylem to leaf water in plants
of the Lake Challa area is 24 ± 28 ‰. Simulating global patterns of leaf water $\delta^2$H enrichment, Kahmen et al. (2013) predicted $\delta^2$H-enrichment to be strong in arid biomes (40-100 ‰), intermediate in temperate biomes (10-30 ‰) and weaker in humid tropical biomes (0-20 ‰). Our values for the semi-arid tropical area of Lake Challa lie in the temperate model. The broad range of $\delta^2$H$_{leaf}$ values measured in this study (from -83 ‰ to 37 ‰) is not surprising as several environmental variables, including ambient temperature, relative humidity,
wind speed and the $\delta^2$H of atmospheric water vapor influence the leaf water deuterium enrichment (Sachse et al., 2012 and references therein). The drier the atmosphere, the windier the conditions and the warmer the air, the larger the rates of transpiration and thus the rates of water uptake (Craig and Gordon, 1965). In our study region, the monthly average temperature varied only slightly (Fig. 2a) and will have a minor effect on the variability of $\delta^2$H$_{leaf}$ and $\varepsilon_{l/x}$ between months. To eliminate additional variability induced by previously reported large diurnal
variations in $\delta^2$H$_{leaf}$ (Cernusak et al., 2002; Li et al., 2006; Kahmen et al., 2008), our samples were taken as much as possible around the same time in the day.

In our study area, leaf phenology appears to exert an important influence on $\delta^2$H$_{leaf}$, and this is also reflected in $\varepsilon_{l/x}$ (Fig. 6). The species-specific variation in $\varepsilon_{l/x}$ is likely explained by differences in plant physiology and
biochemistry. The highest $\delta^2$H$_{leaf}$ and $\varepsilon_{l/x}$ values were displayed by deciduous species, at least on the crater rim and in the savannah. Evergreen plants keeping their foliage during the dry season must be protected against drought stress by a high degree of succulence or sclerophylly (thickened or hardened leaves) to reduce moisture loss (Chabot and Hicks, 1982). Burghardt and Riederer (2003) observed that the peak cuticular transpiration rates of evergreens are approximately one order of magnitude lower than those of deciduous species. Thus, the
adaptive traits of evergreens which reduce water loss and lower transpiration rates result in lower xylem-to-leaf deuterium enrichment. Differences in $\varepsilon_{l/x}$ between habitats are not surprising, as the plants at the lake shore are protected by the crater rim and less transpiration is expected compared to plants in the savannah and on the crater rim.

Extremely low $\varepsilon_{l/x}$ values of respectively -24 ± 30 ‰ and 8 ± 15 ‰ were recorded for *Maerua* sp. and *Thylachium africanum*, two evergreen Capparaceae growing on the crater rim (Fig. 6). This is indicative of very limited evapotranspiration. Slightly depleted $\delta^2$H$_{leaf}$ values relative to $\delta^2$H$_{xylem}$ (i.e. small, negative $\varepsilon_{l/x}$ values) have previously been reported for some trees in *Rhizophora* mangroves (Ladd and Sachs 2015). These authors ascribed this to the high ambient relative humidity, resulting in a small vapor pressure gradient across the leaf
surface (see also Helliker and Ehleringer, 2000; Farquhar et al., 2007). The atypical $\varepsilon_{l/x}$ values of the two Capparaceae in this study are possibly associated with their xerophytic traits, in particular the waxy appearance of the leaves in many species of this family (Elffers et al., 1964). These waxy and leathery leaves are useful

adaptations to survive a long dry season, as plants lose water not only via their stomata but also across the cuticle (Schonherr, 1982). Oliveira et al. (2003) observed that the waxes of *Capparis yco*, a species belonging to the Capparaceae, are very efficient against water loss due to the predominance of *n*-alkanes in their composition. Thus, these waxes on the leaf surfaces of the two Capparaceae species could reduce the plants' transpiration and thus possibly explain their small $\varepsilon_{l/x}$ values. Furthermore, taking into account highly diverse leaf morphology large variations in $\delta^2H_{leaf}$ and $\varepsilon_{l/x}$ between plant species were expected (Smith and Freeman, 2006; Kahmen et al., 2008).

The $\varepsilon_{l/x}$ values of Lake Challa area plants did not show systematic variation according to growth form (that is, trees versus shrubs). Judging from Fig. 6, the plants' habitat did significantly affect $\varepsilon_{l/x}$. However, *Grewia tephrodermis*, *Vepris uguenensis* and *Thylachium africanum* show similar $\varepsilon_{l/x}$ values irrespective of the habitat in which they were sampled (Fig. 6). This suggests that the overall difference in $\varepsilon_{l/x}$ values according to habitat is due to differences in the plant assemblage occurring in each habitat, rather than habitat-specific factors. The temporal variability in $\varepsilon_{l/x}$ was limited with only two of the sampled species (*Sideroxylon* sp. and *Lepisanthes senegalensis*) showing significant differences across seasons. Both species displayed lowest $\varepsilon_{l/x}$ during the long rainy season (SE monsoon) and highest $\varepsilon_{l/x}$ during both the short rainy season (NE monsoon) and the dry season. Several studies observed that stomatal conductance in savannah plants declines during the dry season due to increased vapor pressure deficits and declining soil water availability (Duff et al., 1997; Prior et al., 1997). Surprisingly, O'Grady et al. (1999) detected higher transpiration rates in open-canopy eucalyptus forests in Australia during the dry season than during the wet season, mainly because of higher evaporative demand. Meinzer et al. (1993), on the other hand, found similar mean transpiration rates in a lowland tropical forest tree during the wet and dry seasons despite variation in the leaf-to-air vapor pressure difference. Our data on the majority of species sampled around Lake Challa are consistent with the observations of Meinzer et al. (1993) in that they do not show a significant difference in $\varepsilon_{l/x}$ among seasons. In summary, our results point to the fact that at the local scale of a single study area with several distinct plant habitats, the plant species assemblage and associated prevailing leaf phenology are the primary factors influencing xylem-to-leaf water $\delta^2H$ enrichment, while growth form and seasonality have negligible effects.

Along a major hydroclimate gradient influencing the composition of plant assemblages at the (sub-) continental scale, the $\delta^2H$ of plant leaf wax *n*-alkanes ($\delta^2H_{wax}$) varies with the mean $\delta^2H$ value of local precipitation (Sachse et al., 2004; Huang et al., 2004; Hou et al., 2008; Polissar and Freeman, 2010; Garcin et al., 2012; Tipple and Pagani, 2013); it is this relationship which underpins the use of leaf-wax $\delta^2H$ signatures in hydroclimate reconstruction. Precipitation forms the plant's water source and is supposed to be the primary control of $\delta^2H_{wax}$ (Sachse et al., 2012). However, the relative importance of the potential water sources (precipitation, xylem water, leaf water) for lipid synthesis in plant leaves is unknown. Several authors (Sachse et al., 2004; Feakins and Sessions, 2010; Polissar and Freeman, 2010; Kahmen et al., 2013) stated that the leaf water deuterium enrichment also shapes $\delta^2H_{wax}$. Another constraint for a robust interpretation is the limited understanding of the temporal integration of environmental conditions in $\delta^2H_{wax}$. Finally, the 'net or apparent fractionation' between precipitation and leaf wax *n*-alkanes, which integrate these uncertainties, is used for paleoclimate reconstructions. Despite its enormous potential, hydroclimate interpretations remain troubled by uncertainties in the effects of

past variation in water source $\delta^2$H, xylem-to-leaf $\delta^2$H enrichment, and the biosynthetic isotopic depletion which occurs during *n*-alkane synthesis (Sessions et al., 1999; Liu and Yang, 2008; Smith and Freeman, 2006; Sachse et al., 2012). This study investigated the first two of these sources of uncertainty. The third source of uncertainty requires investigations into whether the effects of growth form, phenology, habitat and seasonality that are (not) reflected in $\varepsilon_{l/x}$, are preserved in the leaf wax *n*-alkanes.

**5 Conclusions**

In this study, we measured $\delta^{18}$O and $\delta^2$H of precipitation, lake water, groundwater and plant xylem and leaf water across different plant species, seasons and habitats with varying distances to Lake Challa in equatorial East Africa. We found that the trajectory of the air masses delivering rain to the area considerably influences the seasonal signature of water isotopes in precipitation, but that not all of its variability can be explained in this way. Lake-surface water showed stable $\delta^{18}$O$_{lake}$ and $\delta^2$H$_{lake}$ with, counterintuitively, seasonally lowest isotopic values during the dry season.

No statistical differences were observed between the source water of evergreen and deciduous plants in the three principal habitats around Lake Challa, as inferred from the intersection point ($\delta^2$H$_{LMWL-int}$) of the plants' LELs with the LMWL. We found that the large seasonal variability in δ²H of precipitation was not reflected in the isotopic composition of xylem water. In all three habitats, the plants' principal source water was NE monsoon precipitation falling during the short rainy season (in this year, mostly November-December), likely because these first rains following the long dry season recharged the dry soil. The plants' available water pool was replenished only stepwise by more enriched precipitation from the SE monsoon falling during the long rainy season (in this year, February-May). Consequently, only a minor temporal shift in the isotopic composition of xylem water was observed. These results are in agreement with the 'two water world' hypothesis, where plants rely on a static water pool while a mobile water pool recharges groundwater and is exported to streams as run-off. The evaporation distance (ED) indicates that spatial variability in water resource use exists in the study region. ED values of trees were higher than shrubs in both the lake shore and crater rim habitat. At the crater rim, this indicates that trees use more topsoil water, presumably because the trees' root distribution is driven by their high nutrient needs to sustain a large canopy. At the lake shore, this indicates that trees take up a larger fraction of lake water compared to shrubs. In contrast to water source, leaf phenology (deciduous versus evergreen) plays a key role in determining the xylem-to-leaf water deuterium enrichment in this semi-arid tropical environment. Deciduous species gave highest $\varepsilon_{l/x}$ values, probably because evergreens are better protected against loss of moisture.

Our observations have important implications for the interpretation of δ²H of plant leaf wax *n*-alkanes from paleohydrological records in tropical East Africa, as hydroclimate interpretations remain troubled by uncertainties in the effects of past variation in water source δ²H and leaf water deuterium enrichment. Future studies should establish whether the interspecies variability in xylem-leaf enrichment (24 ± 28 ‰) has the

565 potential to bias paleoclimate reconstructions, given the floristic diversity and likelihood of changes in species assemblage with climate shifts.

**Acknowledgments**

The authors would like to thank two anonymous referees for valuable comments on a previous version of this manuscript. We are grateful to Caxton Oluseno for sample collection and J.J. Wieringa for help with plant
570 identification. We thank the Special Research Fund of Ghent University for support to L.D.W. (BOF CRA 01GB2312) and the German Research foundation (DFG) for support to A.H. P.H.F. is funded by Programa de Formación de Personal Avanzado CONICYT, BECAS CHILE and by the Bijzonder Onderzoeksfonds (BOF) of Ghent University.

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

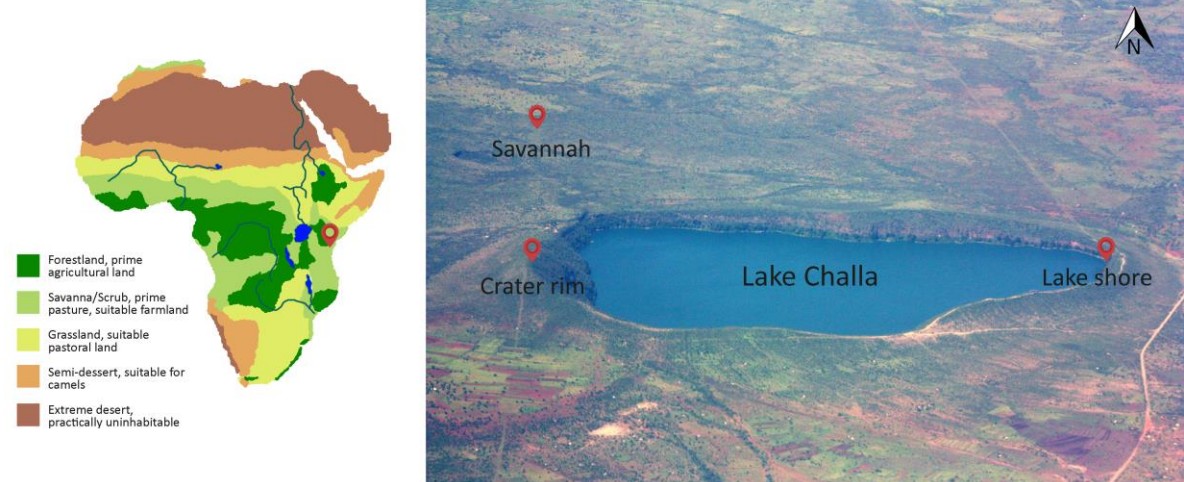

**Figure 1: Lake Challa, situated in equatorial East Africa, on a continent-scale vegetation map (left) with the different sampled habitats (right). Sampling sites in savannah, at the lake shore and on the crater rim are indicated by red dots. Adapted from Wikimedia Commons (2010).**

**Table 1: Studied plant species and family with their respective growth form, leaf phenology and habitat.**

| Species | Family | Growth form | Leaf phenology | Habitat |
|---|---|---|---|---|
| *Acacia gerrardii* | Leguminosae | Tree | Evergreen | Savannah |
| *Boswellia neglecta* | Burseraceae | Tree | Deciduous | Crater rim |
| *Ficus sycomorus* | Moraceae | Tree | Deciduous | Lake shore |
| *Lepisanthes senegalensis* | Sapindaceae | Tree | Evergreen | Lake shore |
| *Sideroxylon sp.* | Sapotaceae | Tree | Evergreen | Lake shore |
| *Commiphora africana* | Burseraceae | Shrub | Deciduous | Crater rim |
| *Euphorbia tirucalli* | Euphorbiaceae | Shrub | Evergreen | Crater rim |
| *Grewia tephrodermis* | Tiliaceae | Shrub | Deciduous | Savannah, crater rim |
| *Maerua sp.* | Capparaceae | Shrub | Evergreen | Crater rim |
| *Thylachium africanum* | Capparaceae | Shrub | Evergreen | Lake shore, crater rim |
| *Vepris uguenensis* | Rutaceae | Shrub | Evergreen | Savannah, crater rim |
| *Ximenia americana* | Olaceae | Shrub | Evergreen | Savannah |
| *Enteropogon macrostachyus\** | Poaceae | Grass | Perennial | Savannah |
| *Themeda triandra\** | Poaceae | Grass | Perennial | Savannah, crater rim |

\* The whole plant was sampled which consisted mainly of green leaves and thus represented leaf water.

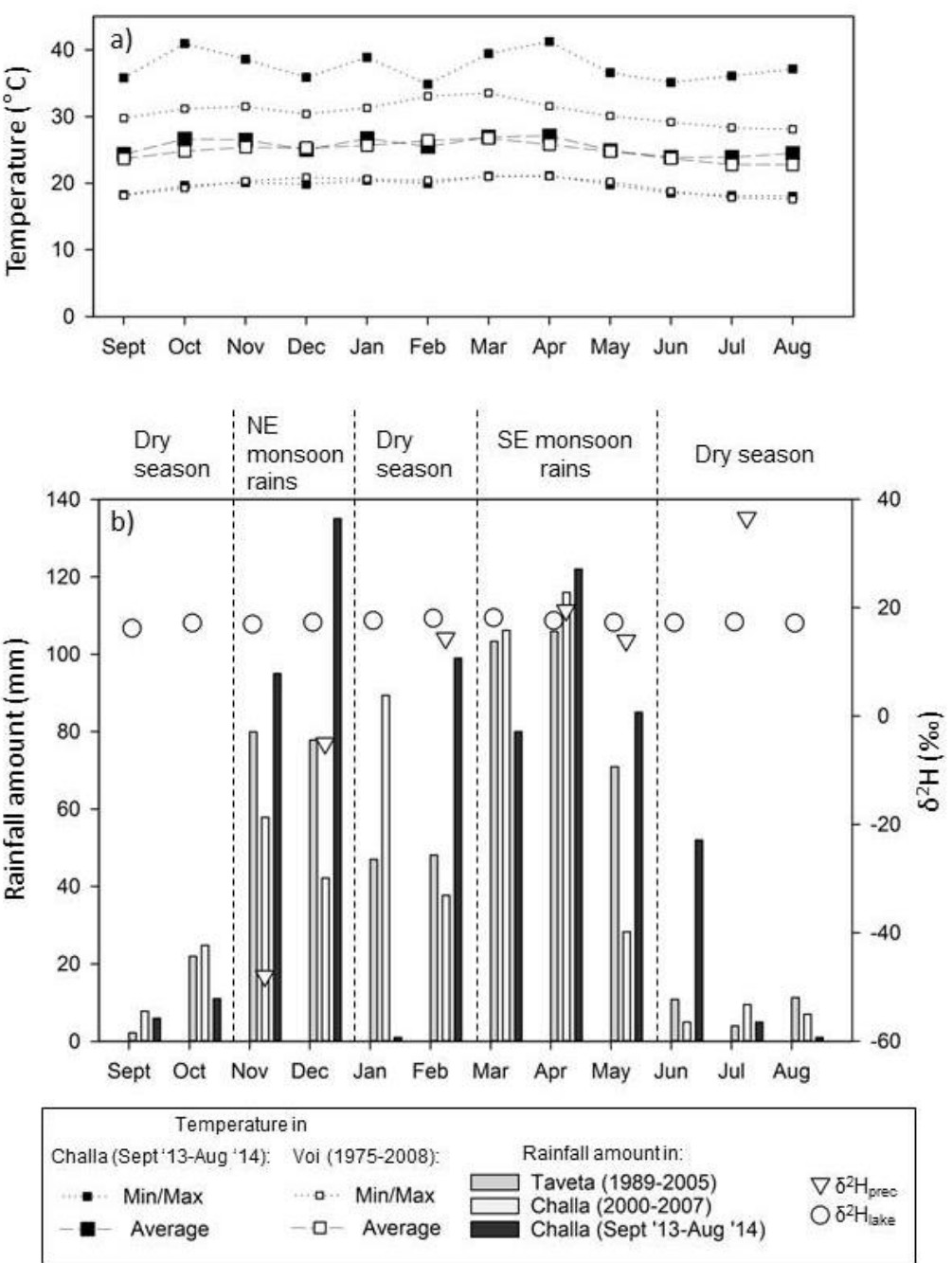


**Figure 2: a) At Lake Challa, the monthly average temperature varied slightly between 23.9 °C in June-July and 27.1 °C in April. The minimum and maximum temperatures for both the study area and nearby Voi (Kenya) are shown. b) Monthly rainfall distribution from September 2013 to August 2014 with the isotopic composition of precipitation and lake water (δ²H$_{prec}$ and δ²H$_{lake}$). The total amount of rainfall during the study period (692 mm) was slightly above**
**reported values for the long-term mean annual precipitation in the Challa region. Note that the 2014 long rainy season atypically already started in February and ceased in June.**

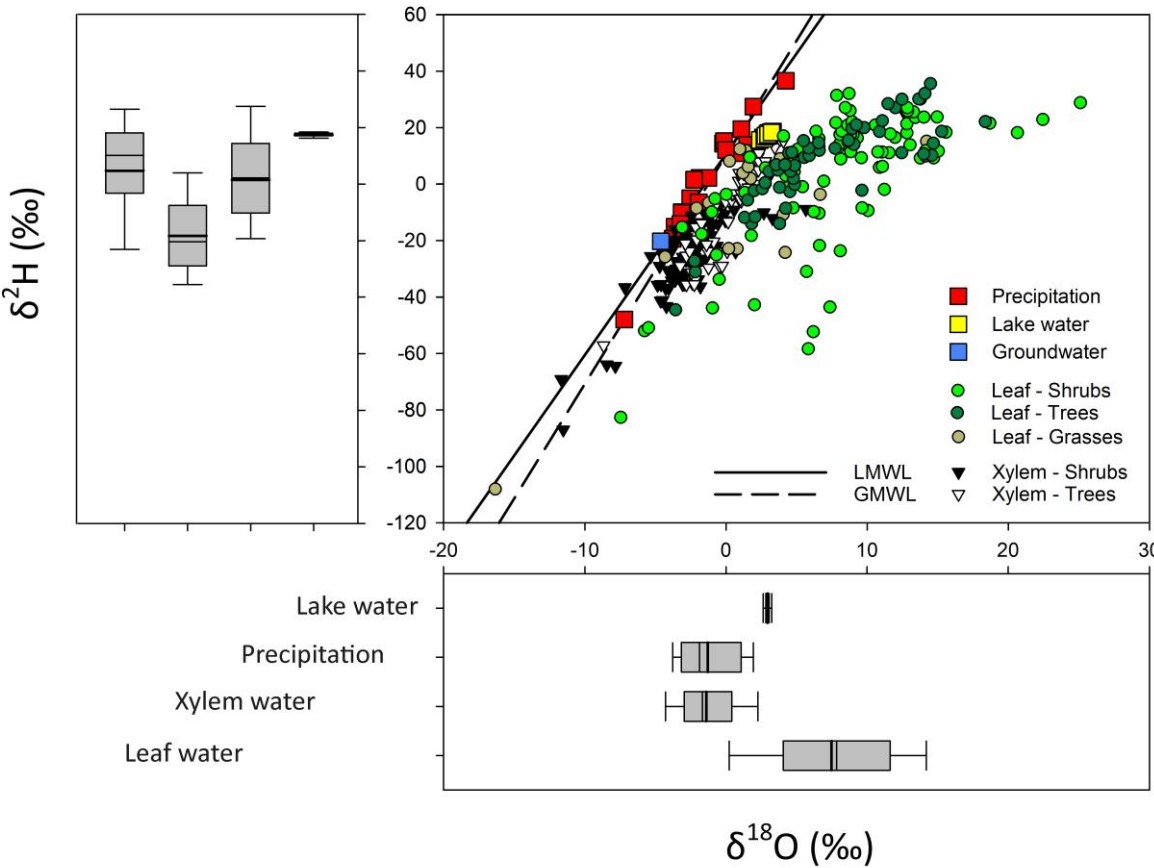

**Figure 3: Xylem and leaf water δ²H and δ¹⁸O values of all plant species, seasons and plant habitats in the vicinity of Lake Challa, and lake, precipitation and groundwater against the LMWL (M²H= 7.12* ¹⁸O + 10.69 ‰, black line). The boxplots show the mean (bold line), minimum, first quartile, median, third quartile and maximum for the isotopic composition of leaf water, xylem water, precipitation and lake water. LMWL: local meteoric water line, GMWL: global meteoric water line (δ²H = 8.1*δ¹⁸O + 10.3 ‰, Rozanski et al., 1993).**


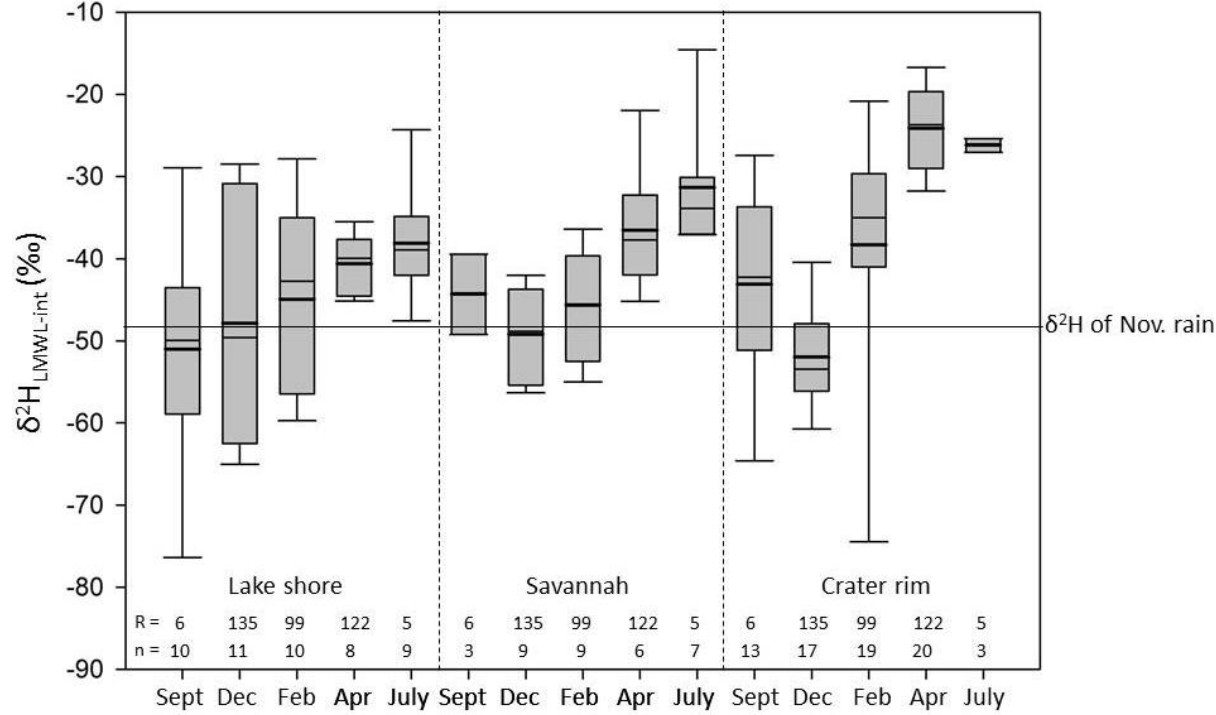


**Figure 4: The average isotopic signature of the source of xylem water (δ²H_LMWL-int), determined from the intersection of xylem water samples with the LMWL, among habitats and seasons. Nov.: November, n: amount of samples, R: rainfall amount (mm).**

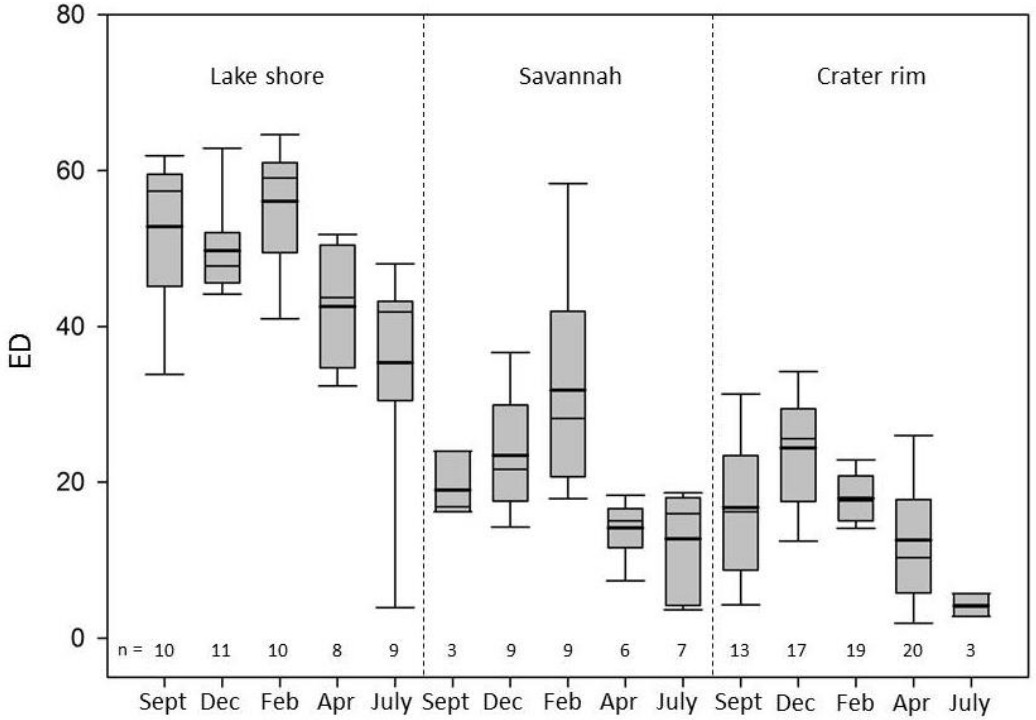


**Figure 5: The evaporation distance (ED) of xylem samples, describing the relative degree of evaporation by calculating the distance from the LMWL along an evaporation line, among habitats and seasons. n: amount of samples.**

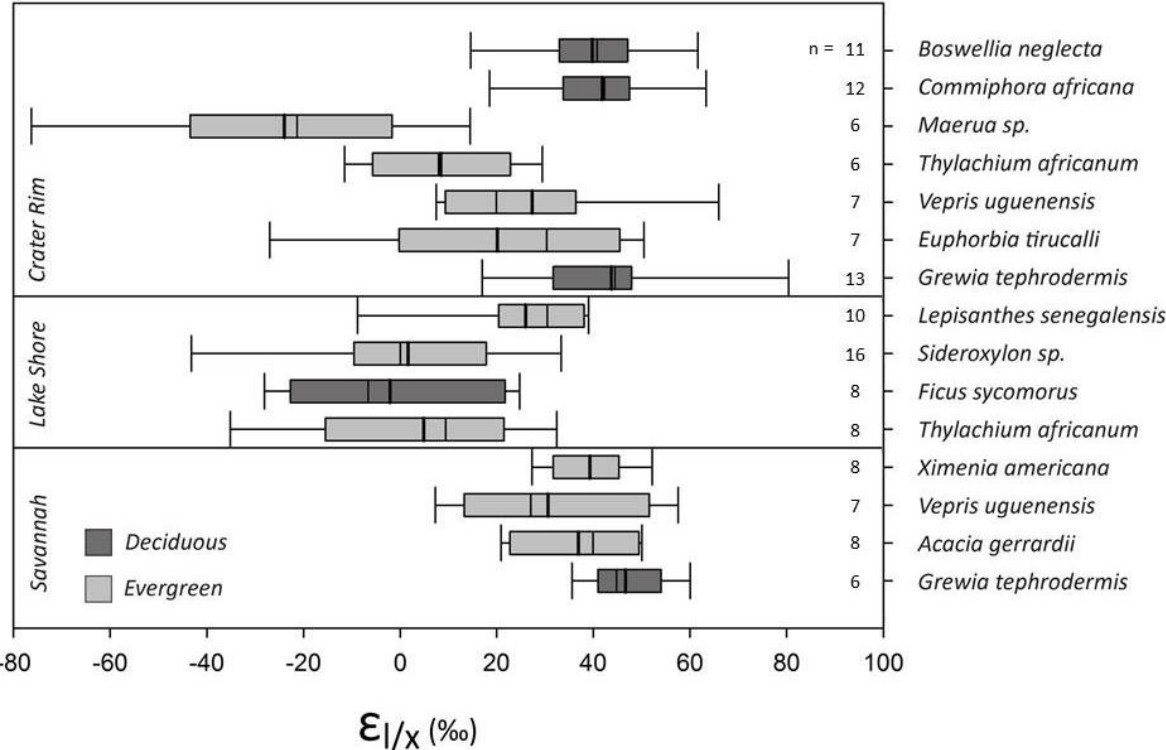

**Figure 6: δ²H enrichment factor e$_{l/x}$, characterizing the hydrogen-isotopic fractionation between xylem and leaf water, among habitat and leaf phenology. Deciduous plants gave higher l$_{l/x}$ than evergreens. Note that *Grewia tephrodermis*, *Vepris uguenensis* and *Thylachium africanum* showed similar a$_{l/x}$ independent from sample habitat. n: amount of samples.**

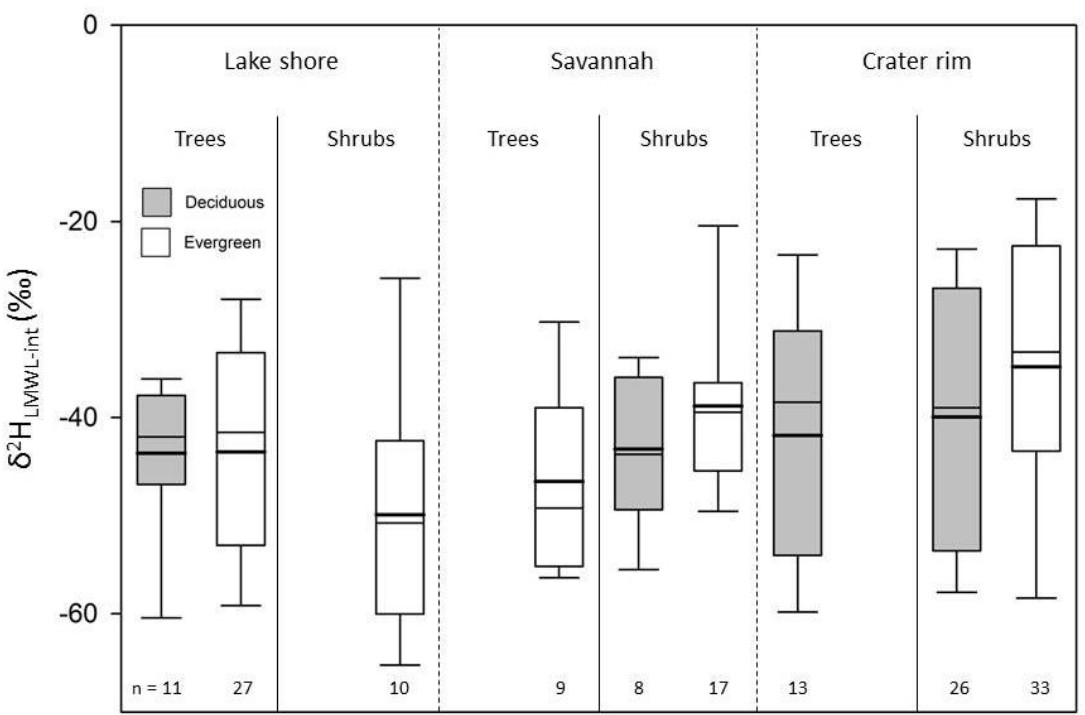

**Figure S1:** $\delta^2H_{LMWL\text{-}int}$ of all sampled plant species, estimating the average isotopic signature of the source of xylem water, among habitat, growth form and leaf phenology. n: amount of samples.

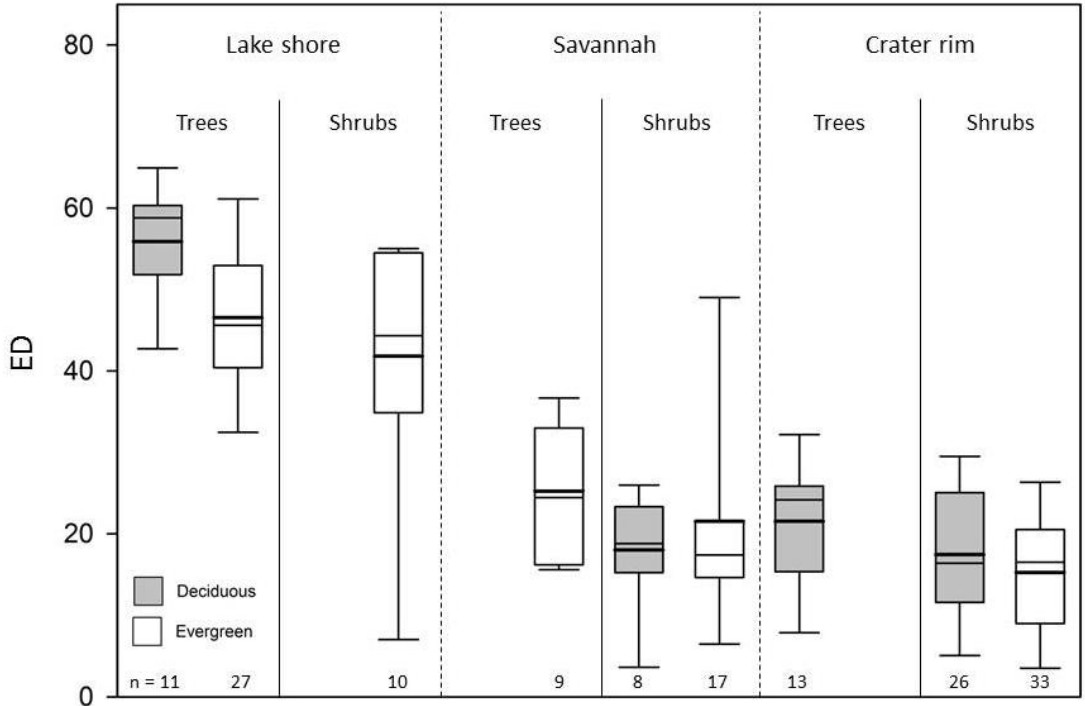


**Figure S2: The evaporation distance (ED) of xylem samples, describing the relative degree of evaporation by calculating the distance from the LMWL along an evaporation line, among habitat, growth form and leaf phenology. n: amount of samples.**