# Peer review of "Plant water resource partitioning and isotopic fractionation during transpiration in a seasonally dry tropical climate"

_Biogeosciences, 2016_

## Referee Comment (RC1) · Anonymous Referee #1 · 30 Sep 2016

GENERAL COMMENTS

The manuscript presents a straightforward and interesting study of deuterium (and 18O) concentrations in leaves, xylem and source water at different habitats of a crater lake at the foothills of Mt. Kilimanjaro. Analyses of heavy isotopes are a potentially very promising tool for eco-hydrological assessments, e.g. regarding the partitioning of soil water among plants. The authors investigate differences in deuterium concentrations among habitats, growth forms, types of leaf phenology and species. The study is also somewhat intended as a baseline study to investigate the potential of heavy isotope analyses of n-alkanes in leaf waxes for paleoclimatic reconstruction. Even though this remains relatively vague throughout the manuscript, the study constitutes an important

contribution of field-measurement based results to the scientific community involved in heavy isotope analysis and its applicability to ecological research questions. The manuscript is well-written and clearly structured, but could partly be improved with regard to language and readability, more methodological detail and references as well as a more clear, overarching message for the scientific community that is to be drawn from the presentation, discussion and conclusion of the presented data.

1. Does the paper address relevant scientific questions within the scope of BG? As an eco-hydrological baseline study aiming to investigate partitioning of source water by analyses of heavy hydrogen and oxygen isotopes, it certainly falls within the scope of BG, covering the field of plant-soil interactions.

2. Does the paper present novel concepts, ideas, tools, or data? The study presents interesting new data from three habitats at the foothills of Mt. Kilimanjaro and gives an outlook of the potential future applicability of isotope analysis in paleoclimatology, i.e. the analysis of heavy isotope concentrations in n-alkanes of leaves to reconstruct past climate, as a relatively novel concept to which the study aims to contribute baseline data.

3. Are substantial conclusions reached? The authors present supporting results for the 'two water world' hypothesis, and they present methodologically sound results on differences in xylem and leaf deuterium concentrations in plants from different habitats, of different growth form and leaf phenology. However, no substantial new insights or overarching conclusions are reached, and the conclusions for the scientific community from these results, e.g. for the paleoclimatic community, could be discussed and delivered in an improved manner.

4. Are the scientific methods and assumptions valid and clearly outlined? The methods are well-described, clearly-structured and feature appropriate references. It some parts, it might be interesting to see a little more current references, if available. In other parts, more detail and references could be added (see details in 'specific comments'

below).

5. Are the results sufficient to support the interpretations and conclusions? The results are described in a very clear way. They do, however, not yield very clear trends, e.g. among habitats, growth forms or leaf phenologies. It is thus difficult to derive strong, overarching conclusions for the scientific community. However, this could be discussed in more detail, if nothing else e.g. regarding what further studies should focus on, and what potential limitations of the study there could have been.

6. Is the description of experiments and calculations sufficiently complete and precise to allow their reproduction by fellow scientists (traceability of results)? Yes, with very few small exceptions (see in 'specific comments' below), the methods seem very solid and reproducible.

7. Do the authors give proper credit to related work and clearly indicate their own new/original contribution? Yes, appropriate references are cited with regard to other studies, and the novel data of the authors is clearly delineable.

8. Does the title clearly reflect the contents of the paper? Yes.

9. Does the abstract provide a concise and complete summary? The abstract gives a complete and concise overview of the results; maybe a final sentence with regard to overarching conclusions/interpretation of these results could be added (also see 'specific comments' below).

10. Is the overall presentation well-structured and clear? The manuscript has a very clear structure that it consistently keeps it each of its parts. It is thus very understandable and pleasant to read.

11. Is the language fluent and precise? Overall, the language is understandable and precise. There are several smaller issues (see 'technical corrections') that could improve readability, e.g. with regard to cutting longer sentences in two, putting commas, or partial rewording. I included several suggestions on how to improve the language

of the manuscript further, some of which might be considered merely a question of style or taste, but others of which should be considered. A final proof-read by a native speaker could also further improve the language.

12. Are mathematical formulae, symbols, abbreviations, and units correctly defined and used? Yes.

13. Should any parts of the paper (text, formulae, figures, tables) be clarified, reduced, combined, or eliminated? There are some small suggestions with regard to clarifying some of the figures and captions (see 'specific comments').

14. Are the number and quality of references appropriate? Yes, references are numerous and their overall quality is appropriate.

15. Is the amount and quality of supplementary material appropriate? There are two interesting supplementary figures.

SPECIFIC COMMENTS

Abstract

Line 28: 'The enrichment in deuterium from xylem water to leaf water averages 24 $\pm$ 28 ‰.' The abstract is very understandable so far. However, this sentence presents a number which, without previous knowledge, is very difficult to place. Perhaps the authors could focus on the interpretation of this value (and its high variability) rather than merely providing it uncommented.

Line 31: The authors later claim that paleoclimatological considerations were one reason for the study. This could be mentioned in a final, somewhat overarching sentence to these conclusions, including the actual implications of the results of the study for the scientific community in general and the paleoclimatological community in particular.

Introduction

Lines 36-38: This sentences could be strengthened with a reference, e.g. if available a

review study on heavy isotope research.

Line 50: The provided reference is quite old (1985), which is not a problem per se, but does make one wonder whether there may be more up to date references that either prove or challenge this assumption?

Line 58: A reference could be added to strengthen this point. Maybe, even a half sentence explaining how evaporation is expected to change the composition could be useful here?

Line 64ff: The authors mention the application of hydrogen isotope geochemistry for paleoclimate research and put some stress on the fact that their study can be considered a baseline study to advance this field. Maybe the reader could profit from a short sentence with a bit more detail or examples of such potential applicability. Furthermore, considering that this seems to be one of the aims of the study, it could use a bit more detail also in the discussion, conclusion and maybe even a summarizing, final sentence in the abstract, that points out the conclusions and implications of this present study for further research and the paleoclimatologic community.

Line 68: 'during which little precipitation falls. . .': Little is a very relative term and could maybe be defined in a bit more detail (e.g. < XX mm/month).

Line 71: Does this limited competition refer to competition among species, or among individual plants? I would try to clarify this a bit more. . .

Line 78: '. . .in several plants. . .': it may sound more interesting to provide a bit more detail here, e.g. ' in XX individuals of XX different species from three distinct habitats. . .'

Line 82: As already suggested for the end of the abstract, a final, overarching sentence focusing of the goals for the scientific community, the novelty of the study, potential implications and applicability for the paleoclimatic community etc might be beneficial to round off this otherwise very solidly formulated and clearly structured introduction.

Methods/Materials

Line 87: It seems that the 'long rains' (2.5 months) are not much longer than the 'short rains' (2 months)... Are these local terms for the seasons? As this could seem confusing, I would recommend sticking to a bit more clear terms, consistently applied throughout the manuscript, e.g. '(long) SE monsoon rains' vs. '(short) NE monsoon rains'.

Line 92: 'annual lake-surface evaporation of c. 1735 mm': How was this value derived/calculated? Or is there a reference?

Line 106: I recommend at a later point (see Figures) to if possible include some rough indications of the distribution of the different land use types in the map, e.g. also different agricultural uses. If this would be possible, the map could be linked in the text here.

Line 120: As lake water seems to only be sampled one single time, it would be good to argue here as to why it would be expected that this is representative, i.e. ground water compositions stable over time.

Line 120: As in the following paragraphs, there seems to not be much detail on sample handling and processing procedures, either in the form of concise descriptions or by providing references and referring to standard procedures.

Lines 126ff: This section could use a bit more detail, e.g. with regard to how the locations were chosen (randomly? arbitrarily?), what considerations were behind and how representative they are for the respective vegetation types. Then, how were the plants within the vegetation types chosen? Were there any further criteria? Also, how was the abundance of species assessed? Merely visually, or were there inventory data at hand, e.g. from previous studies? How common was it that 3 individuals of one species could not be found?

Lines 135ff: In general, the materials and methods (especially this last paragraph) could use a bit more detail and references, particularly with regard to the considerations

behind the applied methods as well as their potential limitations, uncertainties and challenges... Schwendemann et al. (2010: Can deuterium tracing be used for reliably estimating water use of tropical trees and bamboo?) e.g. provide the information that their measurement precision was 2%%.

Line 138: Again, a reference outlining whether this is standard procedure would be helpful. Same for the following sentence: either an explanation on whether this standard protocol, or a reference, or a bit more detail, on how and why and what is important and the considerations behind.

Line 140ff: What kind of leaves were sampled? (randomly? sun exposed? why?) Include reference to standard procedure, or a bit more detail on the considerations behind (if possible also including references).

Line 145f: It feels like this sentence is more more part of sampling/handling than of analysis. . .

Line 151: The authors claim that 'in this paper, isotopic composition is expressed in terms of. . .'. Does this approach differ from other papers? If so, why?

Results

Line 181: mm of precipitation of the dry months would also be interesting here to provide a contrast, as highest values are provided.

Lines 182ff: Precipitation (previous sentences) is of great interest, but the detailed description of temperature is absolutely necessary? Or could this paragraph even be moved to the methods? Is it, particularly the temperature part, essential for later discussions?

Line 188: 'varied by more than 84 ‰' while this absolute value is certainly of interest, it might also be interesting what this means in relative terms (e.g. x-fold variation, or in % normalized my mean, or similar) to give a more simple to grasp impression of the high occurring variability.

Line 193: Are these considerable differences statistically significant? If so, I would mention it.

Line 193: 'In order to draw a reliable LMWL, . . .': would it not have been more reliable if precipitation would have been sampled directly at the study sites? I would remove the word 'reliable'.

Line 204: When did the groundwater composition show these values? Can they be expected to be stable over time? Also see issues addressed in methods.

Lines 230f: Rather than absolute standard deviations (or standard errors, which is it? And among what, days? Months?), it could also be interesting to provide these ranges of temporal variability in relative terms, e.g. by providing the coefficient of variation rather than absolute SD/SE.

Line 233: 'degree of evaporation' of what? A given water sample? How does this parameter describe it (i.e., what do low/high 'values actually mean). At least in the methods, I bit more brief details on the considerations behind and the interpretation of this value might be helpful.

Lines 242ff: 'Finally, among the seven non-grass plant species sampled at the crater rim, a significant difference (p <0.01) was observed between the low ED value for Vepris uguenensis (9 $\pm$ 7 ‰ and the high ED value for Euphorbia tirucalli (25 $\pm$ 7 ‰.' Is this really of significance? Does it tell anything for later, the discussion, conclusions? Or is it merely a significant, but un-interpretable difference that could also be left out?

Line 270: '. . .the ÉŻl/x for $\delta$18 O is not reported here. . .': it is generally not reported much in this study. Why is that? What are the considerations behind? Maybe this could be included in a short sentence in the methods.

Line 277: The provided p-value refers to what exactly? Averages of wet/dry season, or specific months?

Discussion

Line 290: 'other phenomena': For example what kind of phenomena'? Is this in line with other studies?

Line 300: How much drier than usual? Could this be quantified, e.g. in % of long term precipitation?

Lines 305-307: As mentioned before, the (expected) temporal/intra- and inter-annual variability of the groundwater need to be briefly addressed either in the methods or here. Is the singular sampling scheme sufficient? Why?

Lines 336f: Not entirely clear from here, the connections could be made a bit more clear here: similar temporal depletion > similar vertical profile expected. plants with groundwater access > depleted xylem water. But why is the precipitation on Mt Kili-manjaro more depleted? How is the local recharge mechanism in the region? A bit more detail might be of interest.

Line 357: It does not really become clear from this sentence how stem water storage and shallow rooting depth in large trees could be connected.

Line 372: If this precipitation water is retained, then how does it get to the xylem signature? This could be formulated more clearly. As the empty pool is supposed to be refilled annually with the November rains, it also doesn't sound right that it would be retained through several seasons. Needs better explanation.

Line 381: '. . .site at the foot of the crater has deeper soils.' And thus lower fluctuations in soil moisture? What is the connection?

Line 382: '. . .our data seem to confirm. . .': The authors present a quite large data set in this study. What exact data does/does not confirm this hypothesis? All of it?

Line 399ff: The values of the presented study actually seem to be quite distant from arid and lie exactly in temperate... the sentence should thus be rephrased.

Lines 402f: A bit more detail would be helpful, as in how would difference in these

variables be expected to influence seasonality/induce temporal variability in deuterium leaf concentrations? Please discuss a bit more.

Lines 405ff: '. . .highly diverse leaf morphology large variations in $\delta^2$Hleaf between plant species are expected. . .': was this confirmed by the results? Could be sold better if so, has to be discussed if not... This seems to be discussed in the next paragraph, maybe it would be possible to move it their or combine it better...

Line 407: As this is a methodological strength, it could possibly be sold better, e.g. along the lines of 'To eliminate additional variability induced by previously reported large diurnal variations in leaf deuterium, our samples...'

Line 456: 'Despite its enormous potential, hydroclimate interpretations remain troubled by uncertainties 455 in the effects of past variation in water source $\delta^2$H, xylem-to-leaf $\delta^2$H enrichment, and the biosynthetic isotopic depletion which occurs during n-alkane synthesis (Sessions et al., 1999; Liu and Yang, 2008; Smith and Freeman, 2006; Sachse et al., 2012).' A bit more detail, also with regard to more clear conclusions and recommendations for further studies, would be helpful here.

Conclusions

Lines 475ff: Largely a summary and no real, graspable conclusions. Particularly towards the end, add what was learned from the study, what is important for the community, what were the objectives accomplished, and how does this issue need to move on in future studies? This could also come back to the original 3 objectives as mentioned in the abstract: what was specifically learnt for spatial/temporal variability in water resource use, is precipitation composition reflected in the xylem and, lastly, what influences xylem-to-leaf enrichment.

Figures

Line 770: The map on the right side could profit from (approximately) distinguishing land use types/vegetation forms, e.g. with fine lines and also including the lowland

agriculture mentioned in the study site description.

Line 790: Figure 4: Similar to the small numbers (n) provided in the figure, it could potentially be interesting to also provide the average monthly precipitation in mm?

TECHNICAL CORRECTIONS

Introduction

Lines 38-42: The word 'in' should be added twice in this sentence (e.g. fluctuations in the degree of rainout...): 'The hydrogen and oxygen isotopic composition of precipitation varies both spatially and temporally, due to fluctuations i) at the site of evaporation, e.g. in meteorological conditions such as relative humidity (RH), wind and sea surface temperature; and ii) at the site of precipitation, e.g. in the degree of rainout of particular air masses (Craig, 1961; Dansgaard, 1964; Gat 1996; Araguas-Araguas et al., 2000, Gibson et al., 2008).'

Lines 46-47: 'This is in turn used to compare different (sub)surface water bodies with local precipitation (Rozanski et al., 1993; Breitenbach et al., 2010).' Maybe this could be formulated a bit better, e.g. along the lines of 'LMWLs can be used to compare different...'.

Line 49: 'rooting depth' instead of 'root depth'?

Line 50: add comma after bracket: '...1985), so that...'

Line 51: add 'composition of': '...of xylem water represents the composition of the plant water source...'

Line 52: rephrase slightly, e.g.: 'Fractionation during root water uptake has thus far/so far/previously only been found for plants living under xeric conditions...'

Line 53ff: I would suggest to slightly rephrase and cut the sentence in two to enhance readability. E.g.: 'In contrast, the isotopic composition of leaf water differs markedly from that of xylem water. This is because during transpiration in leaves,

lighter molecules diffuse more easily to water vapor than heavier ones.' Or even: '...
This is because transpiration in leaves discriminates towards lighter molecules, while
heavier isotopes (tend to) remain.'?

Line 55f: I would again restructure the sentence slightly to improve readability, e.g.:
'The degree of enrichment (from xylem to leaf) is a function of / mainly depends on temperature, RH and the isotopic composition of the water vapor surrounding the plant...'

Line 62f: This sentence could be modified by adding/changing some words and restructuring a bit, e.g.: 'Consequently, a better understanding of hydrogen fractionation during its incorporation from precipitation over leaf water into plant leaf waxes is needed. The present study...'

Line 68: I would take out the word 'useful' and merely start the sentence with 'Adaptions to...'

Line 70-74: I would restructure and partially reword the sentences slightly to improve readability, e.g.: 'Meinzer et al. (1999) suggested that, at least in pristine dryland ecosystems, competition for water may actually be limited due to pronounced spatial and temporal partitioning of water resources resulting from maximized species diversity. It furthermore appears that the relationship between root biomass in a particular soil layer and contribution of that soil layer to plant water uptake is not always straightforward (Jackson et al., 1995; Stahl et al., 2013).'

Line 75: I would add 'analysis of the' and 'tool': '...Therefore, analysis of the dual stable isotope composition of xylem water could be a valuable tool...'

Line 77: add 'content/fraction/concentration' after '...2H and 18O...'

Line 78: cut into two sentences and provide a bit more detail, e.g.: ,...around Lake Challa. Sampling was carried out/performed monthly/bi-monthly/every x months during successive wet and dry seasons of one complete year.'

Methods/Materials

Line 90: Add 'values/temperature' after '. . .and highest. . .'

Line 93: 'which is mainly derived' sounds a bit strange in this context, maybe replace with e.g. 'which main source is precipitation. . .'

Line 97: add comma: 'a dry forest occurs, with succulents such. . .'

Lines 102ff: I would slightly rephrase, e.g. to: 'The stunted, fruit tree-like appearance of the woody species, mainly Combretaceae, Burseraceae and Anacardiaceae, inspired the first botanists to describe this vegetation formation as 'Obstgartensteppe' ('fruittreegardensteppe'; Volkens, 1897).'

Line 113: 'from entering' instead of 'enter'

Line 114: shorten to '. . ., which could alter isotopic composition of collected water (Friedman et al., 1992).'

Line 117: replace 'monthly' with 'on a monthly basis'.

Line 119: no comma: 'the lake on a monthly basis from. . .'

Line 121: Throughout the manuscript, there are references to the 'rain season' as opposed to the 'dry season'. I believe the correct term would be 'rainy season'? It does appear correctly in other parts of the manuscript. Please use consistently.

Line 139f: reword slightly for better readability: 'In the case of smaller trees and shrubs, a piece of twig was sampled, the outer layer was scraped off using a knife and it was enclosed into sealed vials. . .'

Line 142: change to: 'From grasses, only leaves were sampled.'

Line 154: , change to: '... which, by definition, has H and O concentrations of 0%%...'

Line 154: I would start a new paragraph at the end of line 154, as the topic shifts to the enrichment factor.

Line 156: maybe the (Eq.1) bracket should appear behind the actual equation?

Line 168: add 'i.e. the higher the ratio/proportion/concentration of heavy isotopes.' to the end of the sentence for a bit more clarification?

Line 172: replace 'or' with 'and'

Lines 174f: slightly reword to 'A discussion of the different slopes and intercepts is not scope of this paper, but they were used to calculate $\delta$2175 HLMWL-int.'

Results

Line 182: take out 'slightly'

Line 189: 'sampling period' instead of 'sample period'?

Line 208: as the mean value is actually not shown in the figure, I would move the link to the figure to a different place in the sentence.

Line 208f: As it actually does not vary considerably among all habitats, I would rephrase to: '...$\delta^2$Hxylem varied between plants at the lake shore (-2 $\pm$ 10 ‰ n = 48) and isotopically more enriched plants in the savannah (-25 $\pm$ 12 ‰ p <0.01, n = 34) and on the crater rim (-26 $\pm$ 15 ‰ p <0.001, n = 72).'

Line 210: remove 'Also...', start sentence with 'The...'

Line 214: replace 'caused no significantly different' with 'did not significantly influence 2Hxylem values...'

Line 215: replace 'an effect of season' with 'seasonality in 2Hxylem'?

Line 215: replace 'tree' with 'species' or 'tree species', also in line 218

Line 222: I would move the (Eq. 2 and 3) bracket behind the word 'estimate'

Line 228: replace 'between' with 'among' (more than two), and take out the following word 'the'. I would also replace 'according to' by 'analyzed by'.

Line 233: Split sentence in two: '...degree of evaporation. It is derived...'

Line 237: slightly rephrase to 'Growth form also influenced ED (p<0.05), with lower values for shrubs than...'

Line 241: replace 'however' with 'but'

Line 242: take out 'Finally. . .', start sentence with 'Among. . .'

Line 247: cut sentence in two: '. . .two species of grasses. Across this complete. . .'

Line 248: it should be either 'ranged from. . .to' or 'varied between . . .and. . .'

Line 257: 'had' instead of 'have'?

Line 260: insert the word 'species' after 'shrub' and 'tree'?

Lines 263/264: I would take out the word 'most' in both lines

Lines 264/265: if 'respectively' appears at the end of a sentence, a comma should appear before it ('. . ., respectively.')

Line 269: cut sentence in two: '. . .and $\delta^2$Hleaf values. This yielded an average. . .'

Line 271: technically, the difference is not significant among all habitats (see comment before), so that maybe the sentence could be rephrased, e.g. significant difference between x (. . .) and y and z (. . ., respectively). . .

Discussion

Line 287: the word 'However' should be followed by a comma ('However, . . .').

Lines 294f: rephrase slightly, e.g. to: '. . .(2014), the 2013 rainy season started in mid-November instead of late October and was thus delayed by 2-3 weeks.'

Line 295: Start next sentence with 'Additionally, it already ceased in...'

Line 296: cut sentence in two: '. . .earlier than normal. Rainfall amounts. . .'

Line 297: add comma: 'In addition, the 2014 long. . .'

Line 299: add comma: 'On the other hand, the month...'

Line 299: change to 'main dry season from July to September...'

Line 300: change to 'The $\delta$2Hprec and $\delta$18prec in the dry month of July were clearly more enriched than the corresponding...'

Line 304: remove comma

Line 311: 'very similar': again, a bit of a quantification might be interesting (e.g. within/varied less than 5%? 10%?)

Line 313: the previous sentence states that the seasonality is very small. Now it is described at modest. This should we worded more clear and consistently.

Line 322: I would replace the word 'reduced' with 'similar' or 'similarly small'

Line 323: an additional clarifying half-sentence relating and interpreting this may be helpful, e.g.: '... signature of xylem water, 2Hxylem, when compared to e.g. the seasonality of plants using temporarily more variable surface water.'

Line 343: add comma after bracket: '...), allowing...'

Line 344: add 'e.g.' after 'as'?

Line 344: change to '...in a tropical moist lowland forest in Panama.'

Line 345: change to 'However, in line with our results in a tropical dry lowland...'

Line 345: move 'also' behind bracket

Line 349: change to '...use more topsoil water enriched in heavy isotopes by evaporation.'

Line 349: next sentence, begin with 'In contrast, in the savannah no such...'

Line 351: change to: '...shrubs allow them to access deeper...'

Line 351: next sentence start with 'however': 'However, Meinzer et al...'

Line 352: rephrase to: '...found smaller trees to use deeper sources of water than larger trees, and attributed this to three possible factors.'

Line 353: replace 'first' with 'firstly/secondly/etc'

Line 356: add 'the': '...water content of the topsoil.'

Line 356: add 'storage': '...stem water storage capacity...'

Line 360: 'In this study...' could be confused with the study by Goldstein mentioned in the previous sentence. Maybe better 'In our study...'?

Line 360: put plural: '... evaporative distances...'

Line 361: 'were' instead of 'are'? (make this consistent throughout the manuscript, there still seem to be some small inconsistencies).

Line 363: add 'the' before 'former'

Line 368: add comma after 'Probably, ...'

Line 370: cut sentence in two and rephrase, e.g.: '...distinct 4-month long dry season. They are thus expected to recharge soils to a relatively large degree.'

Line 370: add comma after parantheses: '...(2010), who...'

Line 379: 'was' instead of 'is'? same in following line.

Line 379: add comma after '...mean $\delta$2HLMWL-int, and the compound...' and adjust slightly

Line 380: add 'the': '... than in the savannah.' Continue with comma after 'Probably, ...' in next sentence.

Line 382: add comma after 'hypothesis'.

Lines 391f: adapt slightly to '…high drought tolerance as its leaves wither and die (i.e. become deciduous) under extremely dry conditions while the stem...'

Line 394: delete the word 'is'

Line 404: add 'the': '… account the highly…'

Line 406: replace 'are observed' with 'have been reported'

Line 415: comma after 'Thus, …'

Line 416: remove comma

Lines 416ff: slightly rephrase and cut sentence in two: 'Thus the adaptive traits of evergreens which reduce water loss and lower transpiration rates result in lower xylem-to-leaf deuterium enrichment. Extremely low ÉŻl/x values of respectively -24 $\pm$ 30 ‰ and 8 $\pm$ 15 ‰ were recorded for Maerua sp. and Thylachium africanum, two evergreen Capparaceae growing on the crater rim (Fig. 6). This is indicative of very limited evapotranspiration.'

Line 428: comma after 'Thus, …'

Line 428: replace 'will' with 'could'?

Line 435: take out 'Also…', start sentence with 'The…'

Line 443: add commas: 'Meinzer et al. (1993), on the other hand, found…'

Line 443: cut to '…associated prevailing leaf phenology…'

Line 465: add 'that': '… but that not all…'

Conclusions

Line 466: add 'counterintuitively'? I.e.: '…with, counterintuitively, seasonally lowest isotopic values…'

Figures

Line 770: Figure 1: figure legends of the map on the left side are very small and difficult to read.

Line 770: Figure 1: figure caption: year and citation of Wikimedia?

Line 770: Figure 1: figure caption: expand description of map on the right side a bit? E.g. 'Sampling sites in savannah, at the lake shore and on the crater rim are indicated by red dots.'

Line 770: Table 1: rephrase caption to: 'Studied plant species with their respective growth form, leaf phenology and habitat.'

Line 775: 'Only leaf water sampled': One might wonder why only leaf water was sampled in these species, maybe an explanatory half-sentence could be added for quick understanding when not having read the full article.

Line 780: Temperature a): symbols in this panel are somewhat similar and quite difficult to distinguish... it might be helpful to use a more distinct set of symbols, or work with differently shaded 'corridors' for the sampling period and historic data set, respectively.

Line 780: Rainfall amount b): The information transmitted by this panel is very valuable and clear. However, the 3 different types of precipitation bars in different shadings do make it a bit heavy to look at. I would e.g. suggest trying to color the bars with the highest relevance (i.e. the study period, 2014) black instead of gray (to draw the most focus), and change the color of 89-05 bars to white and of 2000-07 to a light gray.

Line 780: Figure caption: 'already started' instead of 'started already'

Line 790: Figure 4: caption: $\delta 2HLMWL$-int should maybe not be abbreviated in the figure caption

Line 795: commonly, abbreviations (i.e. ED) are annotated on the y-axis. ED does not have any unit?

Figures in general: could it make sense in some of the figures to indicate statistical

differences among habitats/growth forms/leaf phenology/etc by small letters?

---

## Referee Comment (RC2) · Anonymous Referee #2 · 8 Oct 2016

"Plant water resource partitioning and xylem-leaf deuterium enrichment in a seasonally dry tropical climate" by Wispelaere et al. reported isotope data from a data scarce region and used the data to investigate the variations of plant water use both spatially and temporally. The study was carefully conducted and the manuscript is generally well written. I think this would be a valuable contribution to Biogeosciences. At the same time, I think some aspects of the work need to be improved before it could be accepted for publication.

1. The novelty of the study needs to be further emphasized in the Abstract. The objectives and results are clear here, but it reads more like a regional case study. The novelty or importance needs to be emphasized to warrant a publication in an interna-

tional journal.

2. Line 53. This statement requires modification. Based on field observations from a dryland region, a recent study showed that fractionation doesn't occur during root water uptake and it likely occurs during the water redistribution after water uptake. Please refer to Zhao et al. Significant difference in hydrogen isotope composition between xylem and tissue water in Populus euphratica. Plant Cell Environment 2016, for more details.

3. "Study site" section could be incorporated into the "Materials and Methods" section.

4. The sampling time is not clear in the Method section. The non-steady condition in the morning could result in very different isotope signatures of the leaves. More details are needed for the sampling time.

5. Why grass stems were not sampled? It would be a nice comparison between the stem and leaf water isotopic compositions for grasses.

6. The authors used laser spectroscopy method to quantify the isotopic compositions of rainfall, groundwater and plant waters. However, recent studies have showed the potential issues of organic contamination of the spectral signal in the laser spectroscopy method (e.g., West et al. 2010, RCM, 24: 1948-1954, Zhao et al. 2011, RCM 25: 3071-3082). Particularly, Zhao et al. 2011 showed that the isotopic composition differences could be up to 76% for leaf waters between IRMS and laser spectroscopy methods in water-stressed environments. In light of these earlier findings, I think the authors of this study should at least validate some of the leaf water isotope measurements.

7. I like the concept the evaporation distance. If it is cited from others' work, a reference is needed here. Otherwise, the authors should make it clear that "we developed the evaporation distance...". The evaporation distance calculation doesn't seem to be correct. I think the "2H" should be "18O" in the equation. Please double check.

8. There are multiple factors considered in this study, e.g., plant family, growth form,

leaf phenology, habitat, how ANOVA was used for analysis is not clear to me.

9. There are limited rainfall events in the study period and 2014 is an abnormal year. In this context, I think using air trajectories to take a look the source air region of the precipitation could be useful in the interpretation. This reference could be useful in this regard. Soderberg et al. 2013. Using atmospheric trajectories to model the isotopic composition of rainfall in central Kenya. Ecosphere.

10. The plants at the lakeshore produced higher evaporation distance than other two locations. However, the deuterium enrichment from xylem to leaf was smaller for plants at the lake shore compared with other two locations. This is counter-intuitive and needs to be better explained.

Minor comments:

Title: I think "leaf-xylem deuterium enrichment" makes more sense.

Line 91 What is "Voi"?

Line 195 Comparing with global meteoric water line is useful, it would be more meaningful to compare the local meteoric water line with other studies in this region (e.g., Soderberg et al. 2013. Ecosphere).

The "3" and "4" in "C3' and "C4" should be subscripted throughout the manuscript.

---

## Author Comment (AC1) · 5 Nov 2016

GENERAL COMMENTS

The manuscript presents a straightforward and interesting study of deuterium (and 18O) concentrations in leaves, xylem and source water at different habitats of a crater lake at the foothills of Mt. Kilimanjaro. Analyses of heavy isotopes are a potentially very promising tool for eco-hydrological assessments, e.g. regarding the partitioning of soil water among plants. The authors investigate differences in deuterium concentrations among habitats, growth forms, types of leaf phenology and species. The study is also somewhat intended as a baseline study to investigate the potential of heavy isotope analyses of n-alkanes in leaf waxes for paleoclimatic reconstruction. Even though this remains relatively vague throughout the manuscript, the study constitutes an important contribution of field-measurement based results to the scientific community involved in heavy isotope analysis and its applicability to ecological research questions.

**We thank the referee for the time and energy to write a lengthy review, for his/her constructive comments and detailed attention to the manuscript.**

The manuscript is well-written and clearly structured, but could partly be improved with regard to language and readability, more methodological detail and references as well as a more clear, overarching message for the scientific community that is to be drawn from the presentation, discussion and conclusion of the presented data.

**We have reworked the manuscript, trying to increase readability and language and added methodological details and references and tried to clarify the take home message of our manuscript. Further we address your individual comments and suggestions below. Our responses to your comments have been bolded and text added to the manuscript has been italicized here for visual clarity.**

SPECIFIC COMMENTS

Abstract

Line 28: 'The enrichment in deuterium from xylem water to leaf water averages 24 ± 28 ‰·' The abstract is very understandable so far. However, this sentence presents a number which, without previous knowledge, is very difficult to place. Perhaps the authors could focus on the interpretation of this value (and its high variability) rather than merely providing it uncommented.

**Incorporated in comment below.**

Line 31: The authors later claim that paleoclimatological considerations were one reason for the study. This could be mentioned in a final, somewhat overarching sentence to these conclusions, including the actual implications of the results of the study for the scientific community in general and the paleoclimatological community in particular.

**We adapted the closing paragraph of the abstract as follows:**

*According to our results, plant species and their associated leaf phenology are the primary factors influencing the enrichment in deuterium from xylem water to leaf water ($\varepsilon_{l/x}$), with deciduous species giving the highest enrichment; while growth form and season have negligible effects. Our observations have important implications for the interpretation of $\delta^2H$ of plant leaf wax n-alkanes ($\delta^2H_{wax}$) from paleohydrological records in tropical East Africa, given that the temporal variability in the isotopic composition of precipitation is not reflected in xylem water and that leaf water deuterium enrichment is a key factor in shaping $\delta^2H_{wax}$. The large interspecies variability in xylem-leaf enrichment (24 ± 28 ‰) is potentially troublesome, taking into account the likelihood of changes in species assemblage with climate shifts.*

Introduction

Lines 36-38: This sentences could be strengthened with a reference, e.g. if available a review study on heavy isotope research.

**Two references were added: Dansgaard, 1964 and Gat 1996.**

Line 50: The provided reference is quite old (1985), which is not a problem per se, but does make one wonder whether there may be more up to date references that either prove or challenge this assumption?

**Three references were added: Dawson and Ehleringer, 1991 and 1993; Zhao et al., 2006. As well as two references in the sentence above: Zimmermann et al., 1967; Brooks et al., 2010.**

Line 58: A reference could be added to strengthen this point. Maybe, even a half sentence explaining how evaporation is expected to change the composition could be useful here?

**Paragraph amended to:**

*Evaporation directly from the soil causes an isotopic enrichment of soil water available for plant roots (Sachse et al., 2012 and references therein). (…) In contrast, the isotopic composition of leaf water differs markedly from that of xylem water. This is because evaporation discriminates towards lighter isotopologues. As a result, the remaining leaf water after transpiration becomes more enriched in heavy isotopes. The degree of enrichment from xylem to leaf water is a function of temperature, RH and the isotopic composition of water vapor surrounding the plant (Kahmen et al., 2008; Sachse et al., 2012).*

Line 64ff: The authors mention the application of hydrogen isotope geochemistry for paleoclimate research and put some stress on the fact that their study can be considered a baseline study to advance this field. Maybe the reader could profit from a short sentence with a bit more detail or

examples of such potential applicability. Furthermore, considering that this seems to be one of the aims of the study, it could use a bit more detail also in the discussion, conclusion and maybe even a summarizing, final sentence in the abstract, that points out the conclusions and implications of this present study for further research and the paleoclimatologic community.

**The application for paleoclimate research has been further elaborated in the abstract (see above), discussion and conclusions (see below). In the introduction we adapted the paragraph as follows, and divided into 2 paragraphs:**

*As $\delta^2H$ values of leaf wax n-alkanes obtained from lake surface sediments and soils showed tight correlations with $\delta^2H$ values of precipitation (Sachse et al., 2004; Hou et al., 2008; Polissar and Freeman, 2010; Garcin et al., 2012), the $\delta^2H$ signature of n-alkanes, long-chain hydrocarbons with 25-35 carbon atoms derived from fossil plant leaf waxes incorporated in lake sediments, is increasingly being used as paleoclimate proxy (Eglinton and Hamilton, 1967; Mayer and Schwark, 1999; von Grafenstein et al., 1999; Thompson et al., 2003; Tierney et al. 2008; Costa et al. 2014). Consequently, also a better understanding of the hydrogen fractionation is needed which occurs during its incorporation from precipitation via leaf water into plant leaf waxes. The present study is developed in the context of such an application of hydrogen isotope geochemistry for paleoclimate research, focusing on the sediment record of Lake Challa (Verschuren et al. 2009; Barker et al., 2011; Tierney et al., 2011).*

*The study area is located in equatorial East Africa. In this semi-arid tropical region…..*

Line 68: 'during which little precipitation falls. . .': Little is a very relative term and could maybe be defined in a bit more detail (e.g. < XX mm/month).

**Average precipitation amount during dry season (<20 mm/month) is added to the text.**

Line 71: Does this limited competition refer to competition among species, or among individual plants? I would try to clarify this a bit more. . .

**As stated in the text, this limited competition mainly results from a large species diversity. Though, the spatial partitioning of the  water resources has also implications on individuals of the same species.**

*Differential water resource utilization has been shown across different (Ehleringer et al., 1991; Jackson et al., 1999; Goldstein et al., 2008) and within similar growth forms (Field and Dawson, 1998; Meinzer et al., 1999; Stratton et al., 2000).*

Line 78: '. . .in several plants. . .': it may sound more interesting to provide a bit more detail here, e.g. ' in XX individuals of XX different species from three distinct habitats. . .'

*'…in 3 individuals of each of the 14 different species from three distinct habitats…'*

Line 82: As already suggested for the end of the abstract, a final, overarching sentence focusing of the goals for the scientific community, the novelty of the study, potential implications and applicability for the paleoclimatic community etc might be beneficial to round off this otherwise very solidly formulated and clearly structured introduction.

**Following sentence was added to conclude the introduction:**

*The work that we present here reports isotope data from a data scarce region and is intended to provide a basis for the interpretation of leaf wax n-alkane δ²H values applied as (paleo)hydrological proxies.*

Methods/Materials

Line 87: It seems that the 'long rains' (2.5 months) are not much longer than the 'short rains' (2 months)... Are these local terms for the seasons? As this could seem confusing, I would recommend sticking to a bit more clear terms, consistently applied throughout the manuscript, e.g. '(long) SE monsoon rains' vs. '(short) NE monsoon rains'.

**'Long' and 'short' rains are indeed the local terms but in order to avoid confusion we amended the terms in the whole manuscript.**

Line 92: 'annual lake-surface evaporation of c. 1735 mm': How was this value derived/calculated? Or is there a reference?

**The reference (Payne, 1970) is moved from the end of the sentence to directly after the mentioned value of 1735 mm.**

Line 106: I recommend at a later point (see Figures) to if possible include some rough indications of the distribution of the different land use types in the map, e.g. also different agricultural uses. If this would be possible, the map could be linked in the text here.

**See comment below (Figures section).**

Line 120: As ground water seems to only be sampled one single time, it would be good to argue here as to why it would be expected that this is representative, i.e. ground water compositions stable over time.

*We assumed that the isotopic composition of groundwater is seasonally stable, as the groundwater recharge occurs at decadal scale in the region (Zuber, 1983).*

**This sentence has been added to the text.**

Line 120: As in the following paragraphs, there seems to not be much detail on sample handling and processing procedures, either in the form of concise descriptions or by providing references and referring to standard procedures.

**Additional references and information are added (see specific comments below).**

Lines 126ff: This section could use a bit more detail, e.g. with regard to how the locations were chosen (randomly? arbitrarily?), what considerations were behind and how representative they are for the respective vegetation types. Then, how were the plants within the vegetation types chosen? Were there any further criteria? Also, how was the abundance of species assessed? Merely visually,

or were there inventory data at hand, e.g. from previous studies? How common was it that 3 individuals of one species could not be found?

**Three distinct and representative habitats were chosen visually. The abundance of species in each habitat was assessed with the aid of an experienced local guide and each species was sampled in triplicate. The text was adapted as follows:**

*Fourteen plant species with varying growth form (grass, shrub or tree) and leaf phenology (deciduous or evergreen, the latter including succulent) were collected in three distinct habitats (savannah, crater rim, lake shore), representative for the region, around Lake Challa (Fig. 1, Table 1). […] The locally most abundant plant species within each habitat were chosen with the aid of an experienced local guide, although the choice of species was sometimes restricted by practical limitations such as difficulties in reaching certain locations. For each habitat, three individuals of each species were sampled.*

Lines 135ff: In general, the materials and methods (especially this last paragraph) could use a bit more detail and references, particularly with regard to the considerations behind the applied methods as well as their potential limitations, uncertainties and challenges... Schwendemann et al. (2010: Can deuterium tracing be used for reliably estimating water use of tropical trees and bamboo?) e.g. provide the information that their measurement precision was 2‰.

*A variety of water extraction methods for the analysis of the stable isotopes water exist. Cryogenic vacuum extraction of soil pore water remains a challenge, but is an effective method for plant water extractions (Orlowski et al., 2016).*

*The $\delta^2H$ and $\delta^{18}O$ values of water samples were determined using Cavity Ringdown Spectrometry (WS-CRDS, L2120-i, Picarro, USA), coupled with a vaporizing module (A0211 high-precision vaporizer) and a microcombustion module, which eliminates interference of organic compounds (Martín-Goméz et al., 2015). Each sample was measured 10 times, of which the first 5 injections were eliminated in order to overcome memory effects. The measurement uncertainty (± 1σ) of the spectrometer was 0.1 ‰ and 0.4 ‰ for $\delta^{18}O$ and $\delta^2H$, respectively.*

Line 138: Again, a reference outlining whether this is standard procedure would be helpful. Same for the following sentence: either an explanation on whether this standard protocol, or a reference, or a bit more detail, on how and why and what is important and the considerations behind.

*When the plant had a trunk with a diameter of more than 10 cm, a core drill sample (300 mm, diameter 4.30 mm, hard-wood head, Pressler, Recklinghausen, Germany) was extracted, from which the outer layer (epidermis, cortex, bark fibres, and phloem) was removed to prevent contamination with phloem sap.*

Line 140ff: What kind of leaves were sampled? (randomly? sun exposed? why?) Include reference to standard procedure, or a bit more detail on the considerations behind (if possible also including references).

*In case of core sampling, leaves were randomly sampled at different heights and at the four cardinal points, and merged to one bulk sample per replicate to provide samples that were representative for the entire plant. Entire leaves were collected in order to ensure integration of the signal from the entire leaf, given likely isotopic gradients along the length of the leaf (Helliker*

*and Ehleringer, 2000; Sessions, 2006). Leaves were sampled as much as possible between 10 a.m. and 3 p.m. to eliminate additional variability induced by previously reported large diurnal variations in the isotopic composition of leaves (Cernusak et al., 2002; Li et al., 2006; Kahmen et al., 2008).*

Line 145f: It feels like this sentence is more more part of sampling/handling than of analysis. . .

**Moved to sampling section.**

Line 151: The authors claim that 'in this paper, isotopic composition is expressed in terms of. . .'. Does this approach differ from other papers? If so, why?

**There are different ways of expressing isotopic composition, though the way used here is the most commonly used one, therefore the words 'in this paper' are removed.**

Results

Line 181: mm of precipitation of the dry months would also be interesting here to provide a contrast, as highest values are provided.

*'… and lowest in January (short dry season) and August (long dry season) with two times 1 mm of rain (Fig. 2b).'*

Lines 182ff: Precipitation (previous sentences) is of great interest, but the detailed description of temperature is absolutely necessary? Or could this paragraph even be moved to the methods? Is it, particularly the temperature part, essential for later discussions?

**We prefer to keep the temperature description in the result section as it is used for the discussion of the variability of $\delta^2H_{leaf}$. Several environmental variables, including ambient temperature, relative humidity, wind speed and the $\delta^2H$ of atmospheric water vapor influence $\delta^2H_{leaf}$ and thus the leaf water deuterium enrichment ($\varepsilon_{l/x}$). However, in our study region the monthly average temperature varied only slightly and will have a minor effect on the variability of $\delta^2H_{leaf}$ and $\varepsilon_{l/x}$ between months.**

Line 188: 'varied by more than 84 ‰´: while this absolute value is certainly of interest, it might also be interesting what this means in relative terms (e.g. x-fold variation, or in % normalized my mean, or similar) to give a more simple to grasp impression of the high occurring variability.

**The value of 84 ‰ represents the spread (minimum-maximum) in $\delta2H_{prec}$ and not the standard deviation. We now detected that this sentence is otiose, as the sentence that follows gives already the range in $\delta^2H_{prec}$. Thus this sentence is deleted and the next sentence is amended to:**

*The isotopic composition of precipitation is most enriched during the dry month of July with values of 36.6 for $\delta^2H_{prec}$ and 4.2 ‰ for $\delta^{18}O_{prec}$ and most depleted during rainy November with values of -47.9 and -7.2 ‰ (Fig. 2), respectively.*

Line 193: Are these considerable differences statistically significant? If so, I would mention it.

**P-value of 0.08 is added.**

Line 193: 'In order to draw a reliable LMWL, . . .': would it not have been more reliablebif precipitation would have been sampled directly at the study sites? I would remove the word 'reliable'.

**Amended according to reviewers' suggestion.**

Line 204: When did the groundwater composition show these values? Can they be expected to be stable over time? Also see issues addressed in methods.

**Sentence now includes the sampling date. Additional explanation is added in methods, see suggestion above (line 120).**

Lines 230f: Rather than absolute standard deviations (or standard errors, which is it? And among what, days? Months?), it could also be interesting to provide these ranges of temporal variability in relative terms, e.g. by providing the coefficient of variation rather than absolute SD/SE.

**Values given after the mean are always the standard deviations, as mentioned in the first paragraph of the results. We agree that the coefficient of variation can be an additionally interesting parameter. However, instead of sometimes presenting the standard deviation and sometimes the coefficient of variation we prefer to always mention the SD to avoid confusion.**

Line 233: 'degree of evaporation' of what? A given water sample? How does this parameter describe it (i.e., what do low/high values actually mean). At least in the methods, I bit more brief details on the considerations behind and the interpretation of this value might be helpful.

*The evaporation distance (ED, Eq. 4) is a parameter describing the relative degree of evaporation of a xylem water sample. It is derived by calculating the distance of xylem data points from the LMWL along the LEL. The higher this ED value, the further away from the LMWL and the more evaporated the water will be.*

**In the methods:**

*The higher this ED value, the further away from the LMWL and the more evaporated the water will be, i.e. the higher the concentration of heavy isotopes.*

Lines 242ff: 'Finally, among the seven non-grass plant species sampled at the crater rim, a significant difference (p <0.01) was observed between the low ED value for Vepris uguenensis (9 ± 7 ‰ and the high ED value for Euphorbia tirucalli (25 ± 7 ‰.' Is this really of significance? Does it tell anything for later, the discussion, conclusions? Or is it merely a significant, but un-interpretable difference that could also be left out?

**This observation is later used in the discussion related to the C3/CAM metabolism of *Euphorbia tirucalli*. So we decided to keep it.**

Line 270: '. . .the $\acute{E}$ Zl/x for _18 O is not reported here. . .': it is generally not reported much in this study. Why is that? What are the considerations behind? Maybe this could be included in a short sentence in the methods.

**In methods:**

*Both the δ²H and δ¹⁸O values of water samples were measured, but particular focus was given on the δ²H values as this research is intended to provide a basis for the interpretation of leaf wax n-alkane δ²H values as (paleo)hydrological proxies.*

Line 277: The provided p-value refers to what exactly? Averages of wet/dry season, or specific months?

**P-value refers to specific months.**

Discussion

Line 290: 'other phenomena': For example what kind of phenomena'? Is this in line with other studies?

*This indicates that not only the general air mass trajectory but also other phenomena such as different degrees of rainout contributing to the formation of precipitation or temperature and relative humidity control of the initial vapour (Dansgaard, 1964) play an important role in determining the isotopic composition of monthly precipitation in any particular year.*

Line 300: How much drier than usual? Could this be quantified, e.g. in % of long term precipitation?

**This is difficult to quantify and instead of giving a value there is referred to figure 2 for a visual impression.**

Lines 305-307: As mentioned before, the (expected) temporal/intra- and inter-annual variability of the groundwater need to be briefly addressed either in the methods or here. Is the singular sampling scheme sufficient? Why?

**Addressed in suggestion in methods.**

Lines 336f: Not entirely clear from here, the connections could be made a bit more clear here: similar temporal depletion > similar vertical profile expected. plants with groundwater access > depleted xylem water. But why is the precipitation on Mt Kilimanjaro more depleted? How is the local recharge mechanism in the region? A bit more detail might be of interest.

*The distribution of precipitation on Mt. Kilimanjaro changes with elevation with mean annual precipitation (MAP) increasing with altitude along the southern slope to a reach a maximum of ~2700 mm year⁻¹ at 2200 m.a.s.l. and decreasing rapidly further uphill (Hemp, 2001). Maximum groundwater recharge was found at an altitude of ~2,000 m where precipitation is depleted isotopically compared to Lake Challa due to the altitude effect (Dansgaard, 1964).*

Line 357: It does not really become clear from this sentence how stem water storage and shallow rooting depth in large trees could be connected.

**Goldstein et al (1998) explained that the diurnal stem water storage capacity increases exponentially with tree size in tropical forest and that although the amount of water consumed from internal storage is relatively small compared to total daily water uptake from the soil,**

**maximum transpiration rates are maintained for a substantially longer fraction of the day in trees with greater storage capacity.**

*Finally, the larger stem water storage capacity of large trees reduces peak daytime demands for soil water uptake and delay the onset of diurnal leaf water deficits.*

Line 372: If this precipitation water is retained, then how does it get to the xylem signature? This could be formulated more clearly. As the empty pool is supposed to be refilled annually with the November rains, it also doesn't sound right that it would be retained through several seasons. Needs better explanation.

**The 'two water worlds' hypothesis is explained more in detail:**

*[…] This is again in accordance with the 'two water worlds' (TWW) hypothesis of Brooks et al. (2010). This hypothesis challenges the assumption of water being completely mixed in soils and states that the mobile water compartment eventually enters the streams through translatory flow, while the static water compartment consists of initial precipitation that is trapped in soil micropores and remains trapped until the water is used through transpiration by plants in the following dry months (Brooks et al., 2010).*

Line 381: '. . .site at the foot of the crater has deeper soils.' And thus lower fluctuations in soil moisture? What is the connection?

*At the top of the crater rim there was greater variability in monthly mean $\delta^2 H_{LMWL\text{-}int}$, and the compound seasonal trend was more pronounced than in the savannah. Probably, this is because the rim is the driest location with shallow soils on bedrock, while the savannah site at the foot of the crater has deeper soils. Shallow soils have a smaller pool of micropores that can trap water and thus the static water compartment will be more quickly exhausted and replenished with new, isotopically different, water.*

Line 382: '. . .our data seem to confirm. . .': The authors present a quite large data set in this study. What exact data does/does not confirm this hypothesis? All of it?

**'Our data' is replaced by 'the isotopic composition of xylem water seems…'.**

Line 399ff: The values of the presented study actually seem to be quite distant from arid and lie exactly in temperate... the sentence should thus be rephrased.

**Amended according to reviewers' suggestion.**

Lines 402f: A bit more detail would be helpful, as in how would difference in these variables be expected to influence seasonality/induce temporal variability in deuterium leaf concentrations? Please discuss a bit more.

*The drier the atmosphere, the windier the conditions and the warmer the air, the larger the rates of transpiration and thus the rates of water uptake (Craig and Gordon, 1965).*

Lines 405ff: '. . .highly diverse leaf morphology large variations in $\delta^2 H_{leaf}$ between plant species are expected. . .': was this confirmed by the results? Could be sold better if so, has to be discussed if not...

This seems to be discussed in the next paragraph, maybe it would be possible to move it their or combine it better...

**Moved to the end of the next paragraph.**

Line 407: As this is a methodological strength, it could possibly be sold better, e.g. along the lines of 'To eliminate additional variability induced by previously reported large diurnal variations in leaf deuterium, our samples...'

**Amended according to reviewers' suggestion.**

Line 456: 'Despite its enormous potential, hydroclimate interpretations remain troubled by uncertainties in the effects of past variation in water source $\delta^2$H, xylem-to-leaf $\delta^2$H enrichment, and the biosynthetic isotopic depletion which occurs during n-alkane synthesis (Sessions et al., 1999; Liu and Yang, 2008; Smith and Freeman, 2006; Sachse et al., 2012).' A bit more detail, also with regard to more clear conclusions and recommendations for further studies, would be helpful here.

*Precipitation forms the plant's water source and is supposed to be the primary control of $\delta^2H_{wax}$ (Sachse et al., 2012). However, the relative importance of the potential water sources (precipitation, xylem water, leaf water) for lipid synthesis in plant leaves is unknown. Several authors (Sachse et al., 2004; Feakins and Sessions, 2010; Polissar and Freeman, 2010; Kahmen et al., 2013) stated that the leaf water deuterium enrichment also shapes $\delta^2H_{wax}$. Another constraint for a robust interpretation is the limited understanding of the temporal integration of environmental conditions in $\delta^2H_{wax}$. Finally, the 'net or apparent fractionation' between precipitation and leaf wax n-alkanes, which integrate these uncertainties, is used for paleoclimate reconstructions.*

Conclusions

Lines 475ff: Largely a summary and no real, graspable conclusions. Particularly towards the end, add what was learned from the study, what is important for the community, what were the objectives accomplished, and how does this issue need to move on in future studies? This could also come back to the original 3 objectives as mentioned in the abstract: what was specifically learnt for spatial/temporal variability in water resource use, is precipitation composition reflected in the xylem and, lastly, what influences xylem-to-leaf enrichment.

**The conclusion is slightly rewritten with more attention to the original objectives from the abstract/introduction, implications for paleoclimate research and future research:**

*In this study, we measured $\delta^{18}$O and $\delta^2$H of precipitation, lake water, groundwater and plant xylem and leaf water across different plant species, seasons and habitats with varying distances to Lake Challa in equatorial East Africa. We found that the trajectory of the air masses delivering rain to the area considerably influences the seasonal signature of water isotopes in precipitation, but that not all of its variability can be explained in this way. Lake-surface water showed stable $\delta^{18}O_{lake}$ and $\delta^2H_{lake}$ with, counterintuitively, seasonally lowest isotopic values during the dry season.*

*No statistical differences were observed between the source water of evergreen and deciduous plants in the three principal habitats around Lake Challa, as inferred from the intersection point ($\delta^2H_{LMWL-int}$) of the plants' LELs with the LMWL. We found that the large seasonal variability in $\delta^2$H*

*of precipitation was not reflected in the isotopic composition of xylem water. In all three habitats, the plants' principal source water was NE monsoon precipitation falling during the short rainy season (in this year, mostly November-December), likely because these first rains following the long dry season recharged the dry soil. The plants' available water pool was replenished only stepwise by more enriched precipitation from the SE monsoon falling during the long rainy season (in this year, February-May). Consequently, only a minor temporal shift in the isotopic composition of xylem water was observed. These results are in agreement with the 'two water world' hypothesis, where plants rely on a static water pool while a mobile water pool recharges groundwater and is exported to streams as run-off. The evaporation distance (ED) indicates that spatial variability in water resource use exists in the study region. ED values of trees were higher than shrubs in both the lake shore and crater rim habitat, indicating that trees use more topsoil water, presumably because the trees' root distribution is driven by their high nutrient needs to sustain a large canopy. The high ED values of plants at the lake shore further indicate that these plants used a significant fraction of lake water, as expected. Based on our results, leaf phenology (deciduous versus evergreen) plays a key role in determining the xylem-to-leaf water deuterium enrichment in this semi-arid tropical environment. Deciduous species gave highest $\varepsilon_{l/x}$ values, probably because evergreens are better protected against loss of moisture.*

*Our observations have important implications for the interpretation of $\delta^2H$ of plant leaf wax n-alkanes from paleohydrological records in tropical East Africa, as hydroclimate interpretations remain troubled by uncertainties in the effects of past variation in water source $\delta^2H$ and leaf water deuterium enrichment. Future studies should establish whether the interspecies variability in xylem-leaf enrichment (24 ± 28 ‰) has the potential to bias paleoclimate reconstructions, given the floristic diversity and likelihood of changes in species assemblage with climate shifts.*

Figures

Line 770: The map on the right side could profit from (approximately) distinguishing land use types/vegetation forms, e.g. with fine lines and also including the lowland agriculture mentioned in the study site description.

**We think that adding the land use types will make the figure less clear and that the combination of the study site description and figure works the best. The scale of the figure doesn't allow to make a clear distinction in land use types e.g. between the upper slopes of the inner and outer crater rim.**

Line 790: Figure 4: Similar to the small numbers (n) provided in the figure, it could potentially be interesting to also provide the average monthly precipitation in mm?

**We like the idea of providing rainfall variability in the figure. Figure amended according to reviewers' suggestion.**

TECHNICAL CORRECTIONS

**We appreciate your careful review which enabled us to improve the manuscript with regard to language and readability.**

Introduction

Lines 38-42: The word 'in' should be added twice in this sentence (e.g. fluctuations in the degree of rainout: …): 'The hydrogen and oxygen isotopic composition of precipitation varies both spatially and temporally, due to fluctuations i) at the site of evaporation, e.g. in meteorological conditions such as relative humidity (RH), wind and sea surface temperature; and ii) at the site of precipitation, e.g. in the degree of rainout of particular air masses (Craig, 1961; Dansgaard, 1964; Gat 1996; Araguas-Araguas et al., 2000, Gibson et al., 2008).'

**Amended according to reviewers' suggestion.**

Lines 46-47: 'This is in turn used to compare different (sub)surface water bodies with local precipitation (Rozanski et al., 1993; Breitenbach et al., 2010).' Maybe this could be formulated a bit better, e.g. along the lines of 'LMWLs can be used to compare different: …'.

**Amended according to reviewers' suggestion.**

Line 49: 'rooting depth' instead of 'root depth'?

**Amended according to reviewers' suggestion.**

Line 50: add comma after bracket: ': : :1985), so that: : :'

**Amended according to reviewers' suggestion.**

Line 51: add 'composition of': ': : :of xylem water represents the composition of the plant water source: : :'

**Amended according to reviewers' suggestion.**

Line 52: rephrase slightly, e.g.: 'Fractionation during root water uptake has thus far/so far/previously only been found for plants living under xeric conditions: : :'

**Amended according to reviewers' suggestion.**

Line 53ff: I would suggest to slightly rephrase and cut the sentence in two to enhance readability. E.g.: 'In contrast, the isotopic composition of leaf water differs markedly from that of xylem water. This is because during transpiration in leaves, lighter molecules diffuse more easily to water vapor than heavier ones.' Or even: ': : : This is because transpiration in leaves discriminates towards lighter molecules, while heavier isotopes (tend to) remain.'?

**Amended according to reviewers' suggestion.**

Line 55f: I would again restructure the sentence slightly to improve readability, e.g.: 'The degree of enrichment (from xylem to leaf) is a function of / mainly depends on temperature, RH and the isotopic composition of the water vapor surrounding the plant…'

**Amended according to reviewers' suggestion.**

Line 62f: This sentence could be modified by adding/changing some words and restructuring a bit, e.g.: 'Consequently, a better understanding of hydrogen fractionation during its incorporation from precipitation over leaf water into plant leaf waxes is needed. The present study: : :'

**Amended according to reviewers' suggestion.**

Line 68: I would take out the word 'useful' and merely start the sentence with 'Adaptions to: : :'

**Amended according to reviewers' suggestion.**

Line 70-74: I would restructure and partially reword the sentences slightly to improve readability, e.g.: 'Meinzer et al. (1999) suggested that, at least in pristine dryland ecosystems, competition for water may actually be limited due to pronounced spatial and temporal partitioning of water resources resulting from maximized species diversity. It furthermore appears that the relationship between root biomass in a particular soil layer and contribution of that soil layer to plant water uptake is not always straightforward (Jackson et al., 1995; Stahl et al., 2013).'

**Amended according to reviewers' suggestion.**

Line 75: I would add 'analysis of the' and 'tool': ': : :Therefore, analysis of the dual stable isotope composition of xylem water could be a valuable tool: : :'

**Amended according to reviewers' suggestion.**

Line 77: add 'content/fraction/concentration' after ': : :2H and 18O: : :

**Amended according to reviewers' suggestion.**

Line 78: cut into two sentences and provide a bit more detail, e.g.: ,: : :around Lake Challa. Sampling was carried out/performed monthly/bi-monthly/every x months during successive wet and dry seasons of one complete year.'

**Amended according to reviewers' suggestion.**

Methods/Materials

Line 90: Add 'values/temperature' after ': : :and highest: : :'

**Amended according to reviewers' suggestion.**

Line 93: 'which is mainly derived' sounds a bit strange in this context, maybe replace with e.g. 'which main source is precipitation: : :'

**Amended according to reviewers' suggestion.**

Line 97: add comma: 'a dry forest occurs, with succulents such: : :'

**Amended according to reviewers' suggestion.**

Lines 102ff: I would slightly rephrase, e.g. to: 'The stunted, fruit tree-like appearance of the woody species, mainly Combretaceae, Burseraceae and Anacardiaceae, inspired the first botanists to describe this vegetation formation as 'Obstgartensteppe' ('fruittreegardensteppe'; Volkens, 1897).'

**Amended according to reviewers' suggestion.**

Line 113: 'from entering' instead of 'enter'

**Amended according to reviewers' suggestion.**

Line 114: shorten to ': : :, which could alter isotopic composition of collected water (Friedman et al., 1992).'

**Amended according to reviewers' suggestion.**

Line 117: replace 'monthly' with 'on a monthly basis'.

**Amended according to reviewers' suggestion.**

Line 119: no comma: 'the lake on a monthly basis from: : :'

**Amended according to reviewers' suggestion.**

Line 121: Throughout the manuscript, there are references to the 'rain season' as opposed to the 'dry season'. I believe the correct term would be 'rainy season'? It does appear correctly in other parts of the manuscript. Please use consistently

**Amended according to reviewers' suggestion.**

Line 139f: reword slightly for better readability: 'In the case of smaller trees and shrubs, a piece of twig was sampled, the outer layer was scraped off using a knife and it was enclosed into sealed vials: : :'

**Amended according to reviewers' suggestion.**

Line 142: change to: 'From grasses, only leaves were sampled.'

**Amended according to reviewers' suggestion.**

Line 154: , change to: '... which, by definition, has H and O concentrations of 0‰‰...'

**Amended according to reviewers' suggestion.**

Line 154: I would start a new paragraph at the end of line 154, as the topic shifts to the enrichment factor

**Amended according to reviewers' suggestion.**

Line 156: maybe the (Eq.1) bracket should appear behind the actual equation?

**Amended according to reviewers' suggestion.**

Line 168: add 'i.e. the higher the ratio/proportion/concentration of heavy isotopes.' To the end of the sentence for a bit more clarification?

**Amended according to reviewers' suggestion.**

Line 172: replace 'or' with 'and'

**Amended according to reviewers' suggestion.**

Lines 174f: slightly reword to 'A discussion of the different slopes and intercepts is not scope of this paper, but they were used to calculate HLMWL-int.'

**Amended according to reviewers' suggestion.**

Results

Line 182: take out 'slightly'

**Amended according to reviewers' suggestion.**

Line 189: 'sampling period' instead of 'sample period'?

**Amended according to reviewers' suggestion.**

Line 208: as the mean value is actually not shown in the figure, I would move the link to the figure to a different place in the sentence.

**Amended according to reviewers' suggestion.**

Line 208f: As it actually does not vary considerably among all habitats, I would rephrase to: ': : :2Hxylem varied between plants at the lake shore (-2  10 ‰ n = 48) and isotopically more enriched plants in the savannah (-25  12 ‰ p <0.01, n = 34) and on the crater rim (-26 15 ‰ p <0.001, n = 72).'

*'… varied between plants at the lake shore (-2 ± 10 ‰, n = 48) and isotopically more depleted plants in the savannah (-25 ± 12 ‰, p <0.01, n = 34) and on the crater rim…'*

Line 210: remove 'Also: : :', start sentence with 'The: : :'

**Amended according to reviewers' suggestion.**

Line 214: replace 'caused no significantly different' with 'did not significantly influence 2Hxylem values: : :'

**Amended according to reviewers' suggestion.**

Line 215: replace 'an effect of season' with 'seasonality in 2Hxylem'?

**Amended according to reviewers' suggestion.**

Line 215: replace 'tree' with 'species' or 'tree species', also in line 218

**Amended according to reviewers' suggestion.**

Line 222: I would move the (Eq. 2 and 3) bracket behind the word 'estimate'

**Amended according to reviewers' suggestion.**

Line 228: replace 'between' with 'among' (more than two), and take out the following word 'the'. I would also replace 'according to' by 'analyzed by'.

**Amended according to reviewers' suggestion.**

Line 233: Split sentence in two: ': : :degree of evaporation. It is derived: : :'

**Amended according to reviewers' suggestion.**

Line 237: slightly rephrase to 'Growth form also influenced ED (p<0.05), with lower values for shrubs than...'

**Amended according to reviewers' suggestion.**

Line 241: replace 'however' with 'but'

**Amended according to reviewers' suggestion.**

Line 242: take out 'Finally: : :', start sentence with 'Among: : :'

**Amended according to reviewers' suggestion.**

Line 247: cut sentence in two: ': : :two species of grasses. Across this complete: : :'

**Amended according to reviewers' suggestion.**

Line 248: it should be either 'ranged from: : :to' or 'varied between : : :and: : :'

**Amended according to reviewers' suggestion.**

Line 257: 'had' instead of 'have'?

**Amended according to reviewers' suggestion.**

Line 260: insert the word 'species' after 'shrub' and 'tree'?

**We think it reads more fluently without the word 'species' added.**

Lines 263/264: I would take out the word 'most' in both lines

**Amended according to reviewers' suggestion.**

Lines 264/265: if 'respectively' appears at the end of a sentence, a comma should appear before it(': : :, respectively.')

**Amended according to reviewers' suggestion.**

Line 269: cut sentence in two: ': : :and 2Hleaf values. This yielded an average: : :'

**Amended according to reviewers' suggestion.**

Line 271: technically, the difference is not significant among all habitats (see comment before), so that maybe the sentence could be rephrased, e.g. significant difference between x (: : :) and y and z (: : :, respectively): : :

**Amended according to reviewers' suggestion.**

Discussion

Line 287: the word 'However' should be followed by a comma ('However, : : :').

**Amended according to reviewers' suggestion.**

Lines 294f: rephrase slightly, e.g. to: ': : :(2014), the 2013 rainy season started in mid-November instead of late October and was thus delayed by 2-3 weeks.'

**Amended according to reviewers' suggestion.**

Line 295: Start next sentence with 'Additionally, it already ceased in...'

**Amended according to reviewers' suggestion.**

Line 296: cut sentence in two: ': : :earlier than normal. Rainfall amounts: : :'

**Amended according to reviewers' suggestion.**

Line 297: add comma: 'In addition, the 2014 long: : :'

**Amended according to reviewers' suggestion.**

Line 299: add comma: 'On the other hand, the month: : :'

**Amended according to reviewers' suggestion.**

Line 299: change to 'main dry season from July to September: : :'

**Amended according to reviewers' suggestion.**

Line 300: change to 'The 2Hprec and 18prec in the dry month of July were clearly more enriched than the corresponding: : :'

**Amended according to reviewers' suggestion.**

Line 304: remove comma

**Amended according to reviewers' suggestion.**

Line 311: 'very similar': again, a bit of a quantification might be interesting (e.g. within/varied less than 5%? 10%?)

**The actual values of the reference are already reported at the end of the sentence.**

Line 313: the previous sentence states that the seasonality is very small. Now it is described at modest. This should we worded more clear and consistently.

**Amended according to reviewers' suggestion.**

Line 322: I would replace the word 'reduced' with 'similar' or 'similarly small'

**Amended according to reviewers' suggestion.**

Line 323: an additional clarifying half-sentence relating and interpreting this may be helpful, e.g.: ': : : signature of xylem water, 2Hxylem, when compared to e.g. the seasonality of plants using temporarily more variable surface water.'

**Amended according to reviewers' suggestion.**

Line 343: add comma after bracket: ': : :), allowing: : :'

**Amended according to reviewers' suggestion.**

Line 344: add 'e.g.' after 'as'?

**Amended according to reviewers' suggestion.**

Line 344: change to ': : :in a tropical moist lowland forest in Panama.'

**Amended according to reviewers' suggestion.**

Line 345: change to 'However, in line with our results in a tropical dry lowland...'

**Amended according to reviewers' suggestion.**

Line 345: move 'also' behind bracket

**Amended according to reviewers' suggestion.**

Line 349: change to ': : :use more topsoil water enriched in heavy isotopes by evaporation.'

**Amended according to reviewers' suggestion.**

Line 349: next sentence, begin with 'In contrast, in the savannah no such: : :'

**Amended according to reviewers' suggestion.**

Line 351: change to: ': : :shrubs allow them to access deeper: : :'

**Amended according to reviewers' suggestion.**

Line 351: next sentence start with 'however': 'However, Meinzer et al: : :'

**Amended according to reviewers' suggestion.**

Line 352: rephrase to: ': : :found smaller trees to use deeper sources of water than larger trees, and attributed this to three possible factors.'

**Amended according to reviewers' suggestion.**

Line 353: replace 'first' with 'firstly/secondly/etc'

**Amended according to reviewers' suggestion.**

Line 356: add 'the': ': : :water content of the topsoil.'

**Amended according to reviewers' suggestion.**

Line 356: add 'storage': ': : :stem water storage capacity: : :'

**Amended according to reviewers' suggestion.**

Line 360: 'In this study: : :' could be confused with the study by Goldstein mentioned in the previous sentence. Maybe better 'In our study: : :'?

**Amended according to reviewers' suggestion.**

Line 360: put plural: ': : : evaporative distances: : :'

**Amended according to reviewers' suggestion.**

Line 361: 'were' instead of 'are'? (make this consistent throughout the manuscript, there still seem to be some small inconsistencies).

**Amended according to reviewers' suggestion.**

Line 363: add 'the' before 'former'

**Amended according to reviewers' suggestion.**

Line 368: add comma after 'Probably, : : :'

**Amended according to reviewers' suggestion.**

Line 370: cut sentence in two and rephrase, e.g.: ': : :distinct 4-month long dry season. They are thus expected to recharge soils to a relatively large degree.'

**Amended according to reviewers' suggestion.**

Line 370: add comma after parantheses: ': : :(2010), who: : :'

**Amended according to reviewers' suggestion.**

Line 379: 'was' instead of 'is'? same in following line.

**Amended according to reviewers' suggestion.**

Line 379: add comma after ': : :mean  2HLMWL-int, and the compound: : :' and adjust slightly

**Amended according to reviewers' suggestion.**

Line 380: add 'the': ': : : than in the savannah.' Continue with comma after 'Probably, : : :' in next sentence.

**Amended according to reviewers' suggestion.**

Line 382: add comma after 'hypothesis'.

**Amended according to reviewers' suggestion.**

Lines 391f: adapt slightly to ': : :high drought tolerance as its leaves wither and die (i.e. become deciduous) under extremely dry conditions while the stem...'

**Amended according to reviewers' suggestion.**

Line 394: delete the word 'is'

**Amended according to reviewers' suggestion.**

Line 404: add 'the': ': : : account the highly: : :'

**Amended according to reviewers' suggestion.**

Line 406: replace 'are observed' with 'have been reported'

**Amended according to reviewers' suggestion.**

Line 415: comma after 'Thus, : : :'

**Amended according to reviewers' suggestion.**

Line 416: remove comma

**Amended according to reviewers' suggestion.**

Lines 416ff: slightly rephrase and cut sentence in two: 'Thus the adaptive traits of evergreens which reduce water loss and lower transpiration rates result in lower xylem-to-leaf deuterium enrichment. Extremely low εl/x values of respectively -24 ± 30 ‰ and 8 ± 15 ‰ were recorded for Maerua sp. and Thylachium africanum, two evergreen Capparaceae growing on the crater rim (Fig. 6). This is indicative of very limited evapotranspiration.'

**Amended according to reviewers' suggestion.**

Line 428: comma after 'Thus, : : :'

**Amended according to reviewers' suggestion.**

Line 428: replace 'will' with 'could'?

**Amended according to reviewers' suggestion.**

Line 435: take out 'Also: : :', start sentence with 'The: : :'

**Amended according to reviewers' suggestion.**

Line 443: add commas: 'Meinzer et al. (1993), on the other hand, found: : :'

**Amended according to reviewers' suggestion.**

Line 443: cut to ': : :associated prevailing leaf phenology: : :'

**Amended according to reviewers' suggestion.**

Line 465: add 'that': ': : : but that not all: : :'

**Amended according to reviewers' suggestion.**

Conclusions

Line 466: add 'counterintuitively'? I.e.: ': : :with, counterintuitively, seasonally lowest isotopic values: : :'

**Amended according to reviewers' suggestion.**

Figures

Line 770: Figure 1: figure legends of the map on the left side are very small and difficult to read.

**The legend of the figure is enlarged for better readability.**

Line 770: Figure 1: figure caption: year and citation of Wikimedia?

**Amended according to reviewers' suggestion.**

Line 770: Figure 1: figure caption: expand description of map on the right side a bit? E.g. 'Sampling sites in savannah, at the lake shore and on the crater rim are indicated by red dots.'

**Amended according to reviewers' suggestion.**

Line 770: Table 1: rephrase caption to: 'Studied plant species with their respective growth form, leaf phenology and habitat.'

**Amended according to reviewers' suggestion.**

Line 775: 'Only leaf water sampled': One might wonder why only leaf water was sampled in these species, maybe an explanatory half-sentence could be added for quick understanding when not having read the full article.

***The whole plant was sampled which consisted mainly of green leaves and thus represented leaf water.***

Line 780: Temperature a): symbols in this panel are somewhat similar and quite difficult to distinguish... it might be helpful to use a more distinct set of symbols, or work with differently shaded 'corridors' for the sampling period and historic data set, respectively.

**A more distinct set of symbols is used to better distinguish the two datasets.**

Line 780: Rainfall amount b): The information transmitted by this panel is very valuable and clear. However, the 3 different types of precipitation bars in different shadings do make it a bit heavy to look at. I would e.g. suggest trying to color the bars with the highest relevance (i.e. the study period,

2014) black instead of gray (to draw the most focus), and change the color of 89-05 bars to white and of 2000-07 to a light gray.

**A lighter set of grey intensities is used with the darkest color representing the study period.**

Line 780: Figure caption: 'already started' instead of 'started already'

**Amended according to reviewers' suggestion.**

Line 790: Figure 4: caption:  2HLMWL-int should maybe not be abbreviated in the figure caption

*Figure 4: The average isotopic signature of the source of xylem water ($\delta^2 H_{LMWL-int}$), determined from the intersection of xylem water samples with the LMWL, among habitats and seasons. Nov.: November, n: amount of samples, R: rainfall amount (mm).*

Line 795: commonly, abbreviations (i.e. ED) are annotated on the y-axis. ED does not have any unit?

**ED is dimensionless. Y-axis amended according to reviewers' suggestion.**

---

## Author Comment (AC2) · 5 Nov 2016

Plant water resource partitioning and xylem-leaf deuterium enrichment in a seasonally dry tropical climate" by Wispelaere et al. reported isotope data from a data scarce region and used the data to investigate the variations of plant water use both spatially and temporally. The study was carefully conducted and the manuscript is generally well written. I think this would be a valuable contribution to Biogeosciences. At the same time, I think some aspects of the work need to be improved before it could be accepted for publication.

**We thank the reviewer for his/her appreciation of our work and for its careful review and constructive comments and suggestions. We address the reviewer individual comments and suggestions below. Our responses to your comments have been bolded and text added to the manuscript has been italicized here for visual clarity.**

1. The novelty of the study needs to be further emphasized in the Abstract. The objectives and results are clear here, but it reads more like a regional case study. The novelty or importance needs to be emphasized to warrant a publication in an international journal.

**We thank the reviewer for this valuable suggestion. We adapted the Abstract to further emphasize this:**
**"*Our observations have important implications for the interpretation of $\delta^2 H$ of plant leaf wax n-alkanes ($\delta^2 H_{wax}$) from paleohydrological records in tropical East Africa, given that the temporal variability in the isotopic composition of precipitation is not reflected in xylem water and that leaf water deuterium enrichment is a key factor in shaping $\delta^2 H_{wax}$. The large interspecies variability in xylem-leaf enrichment (24 ± 28 ‰) is potentially troublesome, taking into account the likelihood of changes in species assemblage with climate shifts.*"**

2. Line 53. This statement requires modification. Based on field observations from a dryland region, a recent study showed that fractionation doesn't occur during root water uptake and it likely occurs during the water redistribution after water uptake. Please refer to Zhao et al. Significant difference in hydrogen isotope composition between xylem and tissue water in Populus euphratica. Plant Cell Environment 2016, for more details.

**Amended according to reviewers' suggestion.**

3. "Study site" section could be incorporated into the "Materials and Methods" section.

**Amended according to reviewers' suggestion.**

4. The sampling time is not clear in the Method section. The non-steady condition in the morning could result in very different isotope signatures of the leaves. More details are needed for the sampling time.

**We are aware that large diurnal variations in the isotopic composition of leaf water can occur. Therefore, our samples were taken as much as possible between 10 a.m. and 3 p.m., a shorter time span was technically not feasible.**

*"Leaves were sampled between 10 a.m. and 3 p.m. to eliminate additional variability induced by previously reported large diurnal variations in the isotopic composition of leaves (Cernusak et al., 2002; Li et al., 2006; Kahmen et al., 2008)."*

5. Why grass stems were not sampled? It would be a nice comparison between the stem and leaf water isotopic compositions for grasses.

**For grasses, it was difficult to separate xylem and leaf water. Sampling of grasses was done by taking out the whole plant which consisted mainly of green leaves and thus represented leaf water.**

*"The whole plant was sampled which consisted mainly of green leaves and thus represented leaf water."*

6. The authors used laser spectroscopy method to quantify the isotopic compositions of rainfall, groundwater and plant waters. However, recent studies have showed the potential issues of organic contamination of the spectral signal in the laser spectroscopy method (e.g., West et al. 2010, RCM, 24: 1948-1954, Zhao et al. 2011, RCM 25: 3071-3082). Particularly, Zhao et al. 2011 showed that the isotopic composition differences could be up to 76% for leaf waters between IRMS and laser spectroscopy methods in water-stressed environments. In light of these earlier findings, I think the authors of this study should at least validate some of the leaf water isotope measurements.

**We are aware of the potential issues of organic contamination of the spectral signal in laser spectroscopy. Therefore, a microcombustion module, directly connected to a high precision vaporizer, was used in our measuring device to eliminate organic interferences by combusting the organic compounds. The text was adapted to:**

*"The $\delta^2H$ and $\delta^{18}O$ values of water samples were determined using Cavity Ringdown Spectrometry (WS-CRDS, L2120-i, Picarro, USA), coupled with a vaporizing module (A0211 high-precision vaporizer) and a microcombustion module, which eliminates interference of organic compounds (Martín-Goméz et al., 2015)."*

7. I like the concept the evaporation distance. If it is cited from others' work, a reference is needed here. Otherwise, the authors should make it clear that "we developed the evaporation distance...". The evaporation distance calculation doesn't seem to be correct. I think the "2H" should be "18O" in the equation. Please double check.

*"The isotopic signatures of xylem water were further characterized with a parameter describing the relative degree of evaporation. We developed the evaporation distance, defined as ED and calculated as the distance from the LMWL along an evaporation line, scaled to the $\delta^2H$ axis (Eq. 4)."*

**The equation of the evaporation distance is correct, the equation simply measuring the distance to the LMWL, the distance along the $\delta^{18}O$ axis is multiplied by the slope of the LMWL to scale it on to the $\delta^2H$ axis.**

8. There are multiple factors considered in this study, e.g., plant family, growth form, leaf phenology, habitat, how ANOVA was used for analysis is not clear to me.

**Multi-way ANOVA was used to look for interactions between parameters. Based on these interactions, the significance of the separate predictor variables was further tested with one-way analysis of variance (ANOVA) with the aid of R. Tukey post-hoc comparisons were used to further examine differences.**

9. There are limited rainfall events in the study period and 2014 is an abnormal year. In this context, I think using air trajectories to take a look the source air region of the precipitation could be useful in the interpretation. This reference could be useful in this regard. Soderberg et al. 2013. Using atmospheric trajectories to model the isotopic composition of rainfall in central Kenya. Ecosphere.

**We thank the reviewer for this valuable suggestion. We looked at air trajectories with the aid of the HYSPLIT model and added this new information in the manuscript:**

*"The HYSPLIT model (Draxler and Hess 2004), developed by NOAA, confirmed that there is a distinctly different trajectory for precipitation in November and December (northeast) and April, May and July (southeast). To compute air parcel trajectories, the model required data from the NOAA meteorological database, and trajectories were modeled 310 hours backwards in time starting from the end of the respective month."*

10. The plants at the lakeshore produced higher evaporation distance than other two locations. However, the deuterium enrichment from xylem to leaf was smaller for plants at the lake shore compared with other two locations. This is counter-intuitive and needs to be better explained.

**The plants at the lake shore are protected by the crater rim, so less evaporation and transpiration (smaller fractionation between xylem and leaf water) are expected compared to plants in e.g. the savannah. The higher evaporation distance at the shore is related with plants taking up an important fraction of the lake as source water. The lake has a large surface area that can easily evaporate, resulting in more enriched water available for plants and thus higher evaporation distances.**

**The observation that plants at the shore are taking up (enriched) lake water is already explained in section 4.2, while the following sentence is added in section 4.3:**

**"Differences in $\varepsilon_{l/x}$ between habitats are not surprising, as the plants at the lake shore are protected by the rim and less transpiration is expected compared to plants in the savannah and on the crater rim."**

Minor comments:

Title: I think "leaf-xylem deuterium enrichment" makes more sense.

**We understand the confusion, so the title is changed to:**
**"Plant water resource partitioning and isotopic fractionation during transpiration in a seasonally dry tropical climate"**

Line 91 What is "Voi"?

**Voi is a town located 80 km east of Lake Challa.**

Line 195 Comparing with global meteoric water line is useful, it would be more meaningful to compare the local meteoric water line with other studies in this region (e.g., Soderberg et al. 2013. Ecosphere).

**The following line was added to the text:**
*"Compared to the global meteoric water line ($\delta^2H = 8.1*\delta^{18}O + 10.3$ ‰, Rozanski et al., 1993) and the LMWL of central Kenya ($\delta^2H = 8.3*\delta^{18}O + 11.0$ ‰, Soderberg et al., 2013), the LMWL of the study region ($\delta^2H = 7.1*\delta^{18}O + 10.7$ ‰, n = 18) has a slightly lower slope and intermediate intercept (Fig. 3)."*

The "3" and "4" in "C3' and "C4" should be subscripted throughout the manuscript.

**Amended according to reviewers' suggestion.**